# *SERUM*: Simple, Efficient, Robust, and Unifying Marking for Diffusion-based Image Generation

**Jan Kociszewski**\*, **Hubert Jastrzębski**\*, **Tymoteusz Stępkowski**\*, **Filip Manijak**\*,
**Krzysztof Rojek**\*, **Franziska Boenisch**, **Adam Dziedzic**
CISPA Helmholtz Center for Information Security

## Abstract

We propose *SERUM*: an intriguingly *simple* yet highly effective method for marking images generated by diffusion models (DMs). We only add a unique watermark noise to the initial diffusion generation noise and train a lightweight detector to identify the signature of this watermark directly in the images, simplifying and *unifying* the strengths of prior approaches. *SERUM* provides *robustness* against any image augmentations or watermark removal attacks and is extremely *efficient*, all while maintaining negligible impact on image quality. In contrast to prior approaches, which are often only resilient to limited perturbations and incur significant training, injection, and detection costs, our *SERUM* achieves remarkable performance, with the highest true positive rate (TPR) at a 1% false positive rate (FPR) in most scenarios, along with fast injection and detection and low detector training overhead. Its decoupled architecture also seamlessly supports multiple users by embedding individualized watermarks with little interference between the marks. Overall, our method provides a practical solution to mark outputs from DMs and to reliably distinguish generated from natural images. Our code is available at
`https://github.com/Hubizon/SERUM`.

## 1 Introduction

In recent years, generative models have attracted significant attention for their ability to synthesize highly realistic images. Diffusion models (DMs) (Ho et al., 2020), in particular, have emerged as the leading paradigm, achieving state-of-the-art performance across many tasks. However, this same capability to produce images that are nearly indistinguishable from real ones has raised critical concerns: generated content can be misused for malicious purposes, including the creation of deepfakes and copyright infringement (Mirsky & Lee, 2021; Franceschelli & Musolesi, 2022). Additionally, when synthetic images are published online and later scraped into future training datasets, they were shown to degrade the performance of generative models (Alemohammad et al., 2024; Shumailov et al., 2024) and amplify existing biases (Wyllie et al., 2024). Together, these challenges underscore the urgent need for reliable detection of generated content.

Watermarking has emerged as the de facto standard to facilitate such detection. It embeds imperceptible but algorithmically detectable signals into generated images, enabling distinction between generated and original content. Most watermarking methods consist of two components: an injector that embeds the watermark into generated images and a detector that retrieves or verifies the watermark signal. A naive approach is to apply classical content watermarking techniques (Kansal et al., 2012; Cox et al., 1997; Chen & Wornell, 2001) directly to generated images. For example, the watermark currently used in Stable Diffusion (Cox et al., 2008) operates by altering a specific frequency component in the image's Fourier domain (Wen et al., 2023) after generation. However, while being computationally fast, these approaches frequently degrade image quality or can easily be removed, limiting their practical effectiveness.

To overcome this limitation, several recent works have introduced diffusion-specific watermarking strategies that embed the watermarking process into the generative pipeline itself (Wen et al., 2023; Ci et al., 2024; Fernandez et al., 2023; Yang et al., 2024; Li et al., 2025). While these approaches

---

\*Equal contribution. Correspondence to: adam.dziedzic@cispa.de

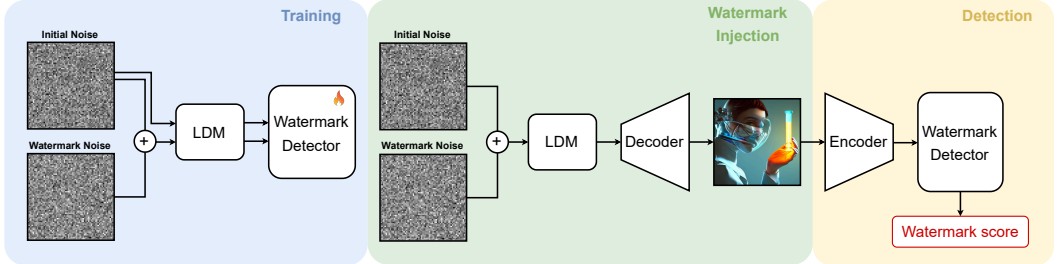

Figure 1: **Overview of *SERUM***. First, we ***train a watermark detector*** to distinguish between watermarked latents (with/without augmentations) from clean latents. Second, to ***inject the SERUM watermark***, the watermark noise is added to the initial random Gaussian noise for diffusion generation and passed through the LDM (Latent Diffusion Model) and decoder to produce a watermarked image. Finally, in order to ***detect the watermark***, the image is encoded using the LDM's encoder and passed through the detector, which outputs a high score for watermarked and a low score for clean images.

significantly enhance watermark robustness, they often incur substantial computational overhead. This overhead typically arises from extensive parameter tuning (Fernandez et al., 2023), or from the reliance on computationally expensive diffusion inversion for watermark detection (Li et al., 2025; Wen et al., 2023; Ci et al., 2024), which in practice leads to delays during the detection process.

As a solution to these drawbacks, we propose *SERUM*, a new DM watermark that injects additional Gaussian noise as a watermark directly into the initial diffusion noise. We train a lightweight external detector to identify the signature of this noise directly in the generated image, effectively eliminating the need for costly diffusion inversion. Our detector also enables flexible training with arbitrary perturbations, and we show that by incorporating augmented images during detector training, we achieve strong watermark robustness to both standard image augmentations and unseen watermark removal attacks. Consequently, *SERUM* combines the benefits of fast injection and detection with the high robustness that results from embedding the watermark into the initial diffusion noise from which the image is generated.

We thoroughly evaluate *SERUM* on latent diffusion models (LDMs), including Stable Diffusion versions 1.4, 2.0, and 2.1 (Rombach et al., 2022), and compare our results to recent watermarking methods such as GaussMarker (Li et al., 2025), Stable Signature (Fernandez et al., 2023), and RingID (Ci et al., 2024). Our experiments demonstrate that *SERUM* consistently achieves superior detection rates under a wide range of image perturbations and dedicated watermark removal attacks, while also maintaining high image generation quality. In addition, we show that the watermark is radioactive (Sablayrolles et al., 2020; Meintz et al., 2025; Kerner et al., 2025), meaning that it remains detectable even in outputs from models fine-tuned on watermarked data. Finally, *SERUM* can be readily extended for multi-user scenarios by assigning distinct noise patterns as user-specific marks and training corresponding classifiers. Taken together, these results highlight the practical utility of *SERUM* for robust data detection in DM.

In summary, we make the following contributions:

- We introduce a novel and efficient method for watermarking generations from DMs, which adds a watermark vector to the initial diffusion noise and trains a lightweight watermark detector.

- Our method seamlessly supports many users by simply instantiating many replicas of our approach per user, while preserving all the desirable properties of the single-instance watermark.

- Extensive experiments on eight perturbations and three leading LDMs show that our method achieves high robustness at low training cost and extremely fast watermark injection and detection.

- Our evaluation on seven advanced watermark removal attack settings demonstrates that *SERUM* achieves state-of-the-art performance, even without explicit training against these attacks.

- Our marking method is highly radioactive: our watermark signal remains detectable even in outputs from models trained or fine-tuned on images watermarked with *SERUM*.

## 2 BACKGROUND AND RELATED WORK

**Diffusion Models (DMs).** DMs (Ho et al., 2020) have emerged as one of the most powerful classes of vision generative models, achieving state-of-the-art results in image synthesis. The central idea of diffusion modeling is to learn the reverse of a gradual noising process that destroys the structure of the training data. At inference, new samples are generated by iteratively denoising an initial Gaussian noise map until a clean image is obtained. While this iterative sampling provides high fidelity, it also incurs significant computational overhead compared to prior models, such as generative adversarial networks. To improve efficiency, Song et al. (2021) introduced the Denoising Diffusion Implicit Model (DDIM), which enables high-quality image generation with fewer denoising steps by employing a deterministic sampling process. Further advances were made with the Latent Diffusion Model (LDM) (Rombach et al., 2022), which integrates a Variational Autoencoder (Kingma & Welling, 2014) to project images into a compressed latent space. Our method and evaluation target LDMs since they represent the state-of-the-art class of image generative models.

**Image Watermarking.** It is indispensable to watermark outputs from vision generative models since recent LDMs (Esser et al., 2024; Podell et al., 2023) are capable of generating photorealistic images and new legislation, for example, the EU Artificial Intelligence (AI) Act (European Parliament and Council of the European Union, 2024) requires us to distinguish between synthetic and authentic content. The existing approaches to watermark LDMs can be broadly categorized into post-processing and in-processing methods. *Post-processing* techniques embed watermarks into generated images after the sampling process (Cox et al., 2008). While extensively studied and well-established, these approaches are vulnerable to removal or degradation. In contrast, *in-processing* methods integrate watermarking directly into the generation pipeline, yielding watermarks that are harder to remove and more seamlessly embedded (An et al., 2024b; Wan et al., 2022). Because of their strong, state-of-the-art performance, we focus on in-processing methods that can be further divided into *tuning-based* and *tuning-free* approaches.

**Stable Signature.** Fernandez et al. (2023) proposed Stable Signature, a state-of-the-art *tuning-based* technique for LDMs. It leverages HiDDeN (Zhu et al., 2018), a framework composed of an encoder, decoder, and adversarial discriminator. The encoder hides a binary message in a cover image, while the discriminator ensures visual indistinguishability. The system is trained to be robust against distortions like cropping and JPEG compression by including noise layers between the encoder and decoder. Stable Signature adapts this by fine-tuning a diffusion decoder to create watermarked images extractable by the HiDDeN decoder. While this allows for fast injection without computational overhead during generation, it relies on the pre-trained HiDDeN extractor, which necessitates substantial training time. Additionally, its reliance on pixel space causes it to generalize poorly to advanced attacks, making it an undesirable choice if robustness is an important metric.

**Tuning-free Watermarks.** To avoid the costly fine-tuning of LDMs, tuning-free methods add a watermark directly to the initial diffusion noise. Tree-Ring (Wen et al., 2023) embeds the watermark in the frequency space of the initial noise. The method achieves a high level of robustness against common image transformations, such as cropping, dilation, and rotation. Unlike the Stable Signature approach, which modifies the decoder, Tree-Ring influences the sampling process itself, allowing for watermark detection by inverting the diffusion process and analyzing the retrieved noise. RingID (Ci et al., 2024) extends Tree-Ring by employing multi-channel heterogeneous watermarking, enhancing its capacity to identify multiple keys, and providing slight improvements in verification performance. Gaussian Shading (Yang et al., 2024) is another tuning-free approach that embeds bits directly into the initial noise of the diffusion model. GaussMarker (Li et al., 2025) ensembles the techniques of both Tree-Ring and Gaussian Shading. It introduces a Gaussian Noise Restorer, which is a trained component designed to improve results against common attacks such as rotation, cropping, and scaling. Although it offers improved performance, it requires additional training. Additionally, like most tuning-free methods, its watermark detection relies on the computationally expensive DDIM inversion. In contrast, our detector operates directly on the generated images. We directly compare *SERUM* to the state-of-the-art tuning-based Stable Signature as well as tuning-free RingID and GaussMarker methods.

**TrustMark.** To build a more resilient post-processing watermarking method, the authors proposed training two separate networks: TrustMark (the watermarking model) and TrustMark-RM (the watermark remover). By jointly training TrustMark to embed watermarks and TrustMark-RM to undo TrustMark's transformations, the system embeds a signal that is difficult to remove using standard

perturbations. While the method performs well against simple transformations, its watermark remains easy to strip using rotations or static noise, highlighting its fundamental limitations.

## 3 OUR *SERUM* WATERMARKING METHOD

*SERUM unifies* the best elements from both worlds of DM watermarking. On the one hand, it matches the speed of tuning-based watermarking methods for both injection and detection, while *training only a lightweight external watermark detector*, thereby leaving the original model intact and substantially reducing training costs. On the other hand, similar to tuning-free approaches, our method *adds the watermark to the initial diffusion noise*, improving robustness. However, in contrast to prior methods, our *SERUM* eliminates the need for computationally expensive DDIM inversion during detection by relying on an external watermark detector. This significantly improves detection speed and adds the advantage that the detector can be further fine-tuned to be robust against any image augmentations or watermark removal attacks.

**Watermark Embedding.** The injection of *SERUM* is relatively simple. Let us denote $\eta \in \mathbb{R}^{c \times w \times h}$ as the initial diffusion noise drawn from the normal unit distribution every time an image is generated ($\eta \sim \mathcal{N}(\mathbf{0}, I)$), where $c$ represents the number of channels, and $w \times h$ are the spatial width and height dimensions, respectively. We inject a watermark noise in the latent space by substituting $\eta$ with $\eta'$:

$$A' = \frac{A - \text{mean}(A)}{\text{std}(A)}$$

$$\eta' = \sqrt{1 - \alpha}\eta + \sqrt{\alpha}A',$$

where $A \in \mathbb{R}^{c \times w \times h}$ is the watermark, $A'$ is the normalized watermark and $\alpha$ is a hyperparameter controlling the balance between watermark's detectability and image diversity. Normalization of $A$ provides a theoretical guarantee of low Kullback-Leibler divergence between the distribution of watermarked and non-watermarked noises, as we prove in Appendix B. This gives our method a more theoretically grounded justification of high image fidelity. To ensure that the DM generates high-quality images even after substituting the initial noise with $\eta'$, we set $A$ once to values drawn from a normal unit distribution: $A \sim \mathcal{N}(\mathbf{0}, I)$ before training the detector. This ensures that most sampled images lie in a region of high unit Gaussian probabilities.

**Watermark Detection.** We detect *SERUM* via a lightweight watermark detector. This module is designed to avoid computationally expensive DDIM inversion. Our detector is denoted as $f : \mathbb{R}^{c \times w \times h} \to [0, 1]$ and is used to verify whether a particular image is watermarked. Note that the input $x$ to $f$ for LDMs is in the latent space (after image embedding), thus, its typical dimensions for SD models are $x \in \mathbb{R}^{4 \times \frac{W}{8} \times \frac{H}{8}}$ (Rombach et al., 2022), with $W$ and $H$ being respectively width and height of the image.

We formulate the loss function $\mathcal{L}$ used to update $f$ at each step as a sum of the loss terms: $\mathcal{L}_w$ (for watermarked images) and $\mathcal{L}_n$ (for non-watermarked images), *i.e.*, $\mathcal{L} = \mathcal{L}_w + \mathcal{L}_n$.

The $\mathcal{L}_w$ term of the loss function is responsible for training the model $f$ to produce high confidence scores when watermark noise is added for image generation:

$$\mathcal{L}_w = \underbrace{-\log f(x^*)}_{\text{marked clean}} \underbrace{-\log f(\mathcal{T}(x^*_{:,:m}))}_{\text{marked transformed}} \underbrace{-\log f(x^*_t)}_{\text{marked precomputed}} ,$$

where $-\log f(x^*)$ serves to identify watermarked images $x^*$. The second sub-term $-\log f(\mathcal{T}(x^*_{:,:m}))$ is for watermarked images perturbed with a random image transformation $\mathcal{T}$ sampled from an *augmentation sampler*. Only $m$ samples are perturbed to reduce the computational overhead of dynamic augmentations where $m$ is a hyperparameter. For the sampler's design, we adapt Prioritized Experience Replay (Schaul et al., 2015). The idea is to make the training procedure target perturbations in which the model performs poorly. This, in turn, strengthens the signal pushing the model into the region of robustness against these difficult augmentations. Our version of the algorithm uses a combination of four techniques: (1) adaptive step sizes for quick likelihood adjustments to changes, (2) probability smoothing to guarantee exploration, (3) a temperature term to balance exploration with exploitation, and (4) clipping to ensure that priority values stay in numerically stable ranges. For more details on the sampler, see Appendix A. The third sub-term $-\log f(x^*_t)$ is for precomputed

augmentations. It allows the model to learn robustness against transformations without much additional computing power. Overall, $\mathcal{L}_w$ is a component of the loss function that trains the model to assign high scores to watermarked images, both clean and perturbed.

The $\mathcal{L}_n$ loss term is analogous to $\mathcal{L}_w$, but designed for samples $x$ generated using a clean random initial noise (no watermark added):

$$\mathcal{L}_n = \underbrace{-\log\big(1 - f(x)\big)}_{\text{clean}} \underbrace{-\log\big(1 - f(\mathcal{T}(x_{:,:m}))\big)}_{\text{transformed}} \underbrace{-\log\big(1 - f(x_t)\big)}_{\text{precomputed}}.$$

To reduce training time, perturbations of both $x^*$ and $x$ are precomputed, resulting in $x_t^*$ and $x_t$. For high-epoch training runs, it is important that, at every training step, $m$ samples be dynamically transformed, as this prevents the model from overfitting. For more details, see Appendix D.

The injection of the watermark noise through a simple weighted sum is motivated by two factors: (1) high detectability by the trained watermark detector and (2) a provably lower Kullback-Leibler Divergence to a unit Gaussian distribution than the method proposed by GaussMarker. Thus, our watermark injection has a lower effect on the distribution of watermarked images, while providing high watermark identification performance.

**Multiple Users.** *SERUM* supports multiple users seamlessly by assigning to each user $i$ a unique subset $S_i \subseteq \{1, \ldots, m\}$ of $k$ normalized noise patterns $\{A_p' : p \in S_i\}$ together with a corresponding user-level detector score $D_i$. The watermark injected for user $i$ is obtained by combining all of the patterns belonging to that user. More precisely, for a clean noise sample $\eta$ we define

$$\eta_i' = \sqrt{1-\alpha}\,\eta + \sqrt{\frac{\alpha}{k}} \sum_{p \in S_i} A_p',$$

Each per-pattern detector is trained using the same procedure as described above but with the corresponding $A_p'$ used as the injected noise. These lightweight detectors are cached after training, and the user-specific detector score is constructed as the product

$$D_i(x) = \prod_{p \in S_i} d_p(x),$$

where $d_p(x)$ is the detector score for pattern $p$. Verification for a user $i$ then reduces to evaluating $D_i(x)$ on the generated content. When computing $D_i(x)$, we memoize the values of $d_p(x)$.

For a set of users $\{1, \ldots, n\}$ (with $n \leq \binom{m}{k}$) the overall watermark score for a sample $x$ is defined as

$$S(x) = \max_p d_p(x),$$

where the prediction is labeled as positive when $S(x) > \tau$ and $\tau$ is the lowest threshold such that $S(x)$ produces a false positive rate of at most 1%.

After verifying that the watermark is embedded within the image, the user associated with the detected watermark is given by

$$\hat{i}(x) = \arg\max_i D_i(x).$$

Consequently, the time complexity of training models for multiuser detection is $O(\sqrt[k]{n})$.

## 4 EMPIRICAL EVALUATION

We evaluate *SERUM* along three axes: (1) robustness to standard perturbations and watermark removal attacks, (2) the quality of the generated images, and (3) the impact of watermark strength on both robustness and image quality. Finally, we also assess the multi-user setup. We begin by detailing the experimental setup used for our evaluations.

### 4.1 EXPERIMENTAL SETUP

**Models.** We evaluate our method and the baselines on three versions of Stable Diffusion (SD): 1.4, 2.0, and 2.1 (Rombach et al., 2022). All models are text-to-image and generate images at a resolution

Table 1: **Watermark robustness against perturbations.** Results are calculated on 10,000 samples (5,000 watermarked and non-watermarked) and TPR @ 1% FPR thresholds are calculated separately for each perturbation.

| Method | Model | Perturbation Robustness (TPR @ 1% FPR) | | | | | | | | | |
|---|---|---|---|---|---|---|---|---|---|---|---|
| | | Average | Clean | Rotate | JPEG | C&S | R. Drop | Blur | S. Noise | G. Noise | Bright |
| TrustMark | SD 2.1 | 59.43 | **100.00** | 4.30 | 93.18 | **99.98** | 0.42 | **100.00** | 14.80 | 86.96 | 35.20 |
| | SD 2.0 | 48.22 | 93.28 | 1.88 | 52.20 | 96.00 | 0.14 | 93.14 | 21.14 | 53.58 | 22.58 |
| | SD 1.4 | 59.32 | **100.00** | 4.64 | 93.36 | **100.00** | 0.40 | **100.00** | 14.92 | 86.08 | 34.54 |
| Stable Signature | SD 2.1 | 90.94 | 99.96 | 94.62 | 91.14 | 99.96 | 99.06 | 99.40 | 91.02 | 49.02 | 94.24 |
| | SD 2.0 | 91.09 | 99.96 | 95.76 | 90.96 | **99.92** | 99.18 | 99.52 | 90.80 | 48.36 | 95.36 |
| | SD 1.4 | 90.98 | 99.98 | 95.12 | 89.00 | 99.98 | 98.58 | 99.68 | 90.26 | 51.98 | 94.24 |
| RingID | SD 2.1 | 88.48 | **100.00** | **99.88** | **99.98** | 5.50 | 99.96 | **100.00** | **100.00** | 92.30 | 98.74 |
| | SD 2.0 | 88.64 | **100.00** | **99.88** | **100.00** | 6.02 | 99.98 | **100.00** | **100.00** | 92.66 | 99.20 |
| | SD 1.4 | 88.77 | **100.00** | **99.88** | **100.00** | 5.96 | 99.96 | **100.00** | **100.00** | 93.32 | **99.82** |
| GaussMarker | SD 2.1 | 97.87 | **100.00** | 98.94 | 99.48 | 88.72 | **100.00** | 97.72 | 99.90 | 97.80 | 98.24 |
| | SD 2.0 | 97.78 | **100.00** | 99.00 | 98.92 | 89.08 | **100.00** | 98.02 | 99.94 | 96.16 | 98.86 |
| | SD 1.4 | 97.07 | **100.00** | 98.90 | 97.70 | 85.94 | **99.98** | 95.32 | 99.92 | 96.70 | 99.14 |
| ***SERUM* (Ours)** | SD 2.1 | **99.75** | **100.00** | 99.34 | **99.98** | 99.54 | 99.86 | **100.00** | **100.00** | **99.30** | 99.72 |
| | SD 2.0 | **99.78** | **100.00** | 99.42 | 99.88 | 99.50 | 99.84 | **100.00** | **100.00** | **99.54** | 99.84 |
| | SD 1.4 | **99.76** | **100.00** | 99.26 | **100.00** | 99.58 | 99.90 | **100.00** | 99.98 | **99.46** | 99.68 |

of $512 \times 512$ pixels, with a latent space size of $4 \times 64 \times 64$. Sampling is performed using 50 denoising steps with a guidance scale of 7.5. For CLIP score calculations, we employ the ViT-B/32 model from OpenAI's CLIP framework Radford et al. (2021), which utilizes a Vision Transformer (Dosovitskiy et al., 2021) architecture with a patch size of 32 pixels.

**Datasets.** To train our watermark detector, we use generated images based on the first 40,000 prompts from the *Gustavosta/Stable-Diffusion-Prompts* repository, using the same latent diffusion model (LDM) on which our method is later evaluated. For the TPR@1%FPR evaluation, we use a disjoint test set consisting of 5,000 prompts from the same repository, yielding 10,000 images in total (one clean and one watermarked image per prompt). To assess the perceptual quality of the clean and watermarked images, we additionally compute the Fréchet Inception Distance (FID) using 10,000 images from the COCO 2014 validation split (Lin et al., 2014). In both evaluations, the positive prompt drives the image generation, whereas the negative prompt is kept empty, serving only as a neutral input that contributes no suppressive guidance.

**Baselines.** We compare our method primarily against Stable Signature (Fernandez et al., 2023), TrustMark (Bui et al., 2025), RingID (Ci et al., 2024), and GaussMarker (Li et al., 2025) as these methods yield state-of-the-art results in DM watermarking. For Stable Signature, we adapt the SD decoder using the watermark extractor released in the corresponding repository. This extractor was trained with a wider range of image augmentations, including blur and rotations, which improves robustness against such attacks, although at the expense of a slight reduction in image quality. For our experiments, we fine-tune the decoder on a 2,000-image subset of COCO (Lin et al., 2014), with default parameters. GaussMarker combines multiple strategies and reports state-of-the-art robustness when measured as mean TPR @ 1% FPR across various perturbations, making it a strong reference point. We trained, generated, and evaluated GaussMarker using a modified version of their open source pipeline. For more details, see Appendix H.1.

**Perturbations.** To evaluate robustness, we test *SERUM* under eight different perturbations:

- **Rotation:** Random rotation in the range of $[-90°, 90°]$ with gray padding for corner filling.
- **JPEG Compression:** Quality factor of 25.
- **Crop & Scale (C&S):** Random crop retaining 75% of the original area with an aspect ratio of 1.
- **Random Drop (R. Drop):** Random removal of 64% of the image pixels.
- **Salt & Pepper Noise (S. Noise):** Corruption of 5% of the pixels.
- **Gaussian Noise (G. Noise):** Additive white Gaussian noise with a standard deviation of $\sigma = 0.1$.
- **Brightness:** Application of color jitter by adjusting the brightness with a factor of 6.

Table 2: **Robustness against advanced generative removal attacks.** We report TPR @ 1% FPR evaluated on 10,000 samples.

| Method | Average | VAE | Regen-12 | Rinse-4x8 | Regen-30 | Rinse-2x25 | CtrlRegen | I2V |
|---|---|---|---|---|---|---|---|---|
| TrustMark | 12.30 | 57.62 | 1.32 | 0.70 | 1.06 | 1.02 | 1.18 | 23.20 |
| Stable Signature | 1.18 | 2.70 | 1.30 | 0.62 | 0.42 | 0.76 | 0.86 | 1.60 |
| RingID | 65.40 | **99.98** | 98.22 | 84.12 | 32.58 | 24.88 | 77.82 | 40.20 |
| GaussMarker | 58.01 | 98.36 | 89.30 | 65.80 | 9.72 | 10.40 | 91.06 | 41.40 |
| *SERUM* (Ours) | **90.87** | 99.88 | **99.72** | **99.38** | **88.50** | **90.76** | **99.64** | **58.20** |

- **Gaussian Blur (Blur):** $15 \times 15$ kernel with standard deviation $\sigma$ sampled uniformly from $[0.1, 2.0]$.

We provide additional visualization of the above perturbations in Figure 7.

**Advanced Attacks.** For advanced watermark removal attacks, we follow the WAVES (An et al., 2024a) benchmark and evaluate five types of attacks: VAE, Regeneration (Zhao et al., 2024), Rinse, CtrlRegen (Liu et al., 2025) and Image-to-Video (I2V) (Lu et al., 2025). Details about the exact experimental setup of the attacks are available in Appendix C.

**Hyperparameters.** We train our watermark detector for 50 epochs using Adam ($lr = 10^{-3}, \beta_1 = 0.9, \beta_2 = 0.999$) and a `ReduceLROnPlateau` scheduler (factor $0.2$, patience $2$). The watermark strength is configured with $\alpha = 0.5$. Each training batch consists of 32 clean and 32 watermarked images. The perturbation size $m$ is set to $4$. Before training, we generate 15,000 watermarked images ($x^*$) and their random perturbations.

## 4.2 ROBUSTNESS TO IMAGE PERTURBATIONS

Table 1 presents the performance of *SERUM* under various image perturbations, measured as True Positive Rate (TPR) at a fixed 1% False Positive Rate (FPR). *SERUM* consistently outperforms prior approaches across all tested models and perturbations. Across all model versions, *SERUM* achieves an average TPR above 99.7%, setting a new state-of-the-art in robustness. Specifically, our method maintains near-perfect detection on clean images and demonstrates exceptional resilience to a wide range of augmentations.

In comparison, while TrustMark is resilient to C&S, it exhibits severe vulnerability to rotation, random drop (dropping below 5% TPR), and noise-based perturbations. Similarly, Stable Signature performs well on clean images and C&S, but suffers substantial drops under Gaussian noise (*e.g.,*, 48–52% TPR) and degrades under JPEG compression and rotation. RingID achieves high detection rates for most perturbations but collapses under C&S (5–6% TPR), indicating a critical limitation. Finally, GaussMarker obtains the second-highest average TPR but displays variable performance, specifically lagging in C&S robustness (85–90% TPR). Overall, *SERUM* achieves the highest average TPR and the best stability among all evaluated methods.

## 4.3 ROBUSTNESS TO WATERMARK REMOVAL ATTACKS

**Watermark Removal Attacks.** We further assess robustness of our *SERUM* watermark against the baselines under dedicated watermark removal attacks in Table 2. We report the TPR@1%FPR on 10,000 samples (2,000 for I2V) with our watermark detector not trained on these attacks. It maintains a TPR exceeding 99% across VAE, Regen-12, Rinse-4x8, and CtrlRegen. Even under significantly more aggressive strategies, *SERUM* retains robust performance, yielding TPRs of 90.76% on Rinse-2x25 and 88.50% on Regen-30. Finally, against the most destructive I2V attack, it achieves 58.20%, substantially outperforming the best available competing methods. This displays *SERUM*'s strong ability to generalize to unseen attacks which we further discuss in Appendix E.

We observe a sharp performance dichotomy in competing methods based on the attack intensity. While baselines like RingID and GaussMarker remain effective on low-noise reconstructions (e.g., VAE, Regen-12), their detection rates collapse on high-noise variants like Regen-30 and Rinse-2x25. In contrast, *SERUM* exhibits consistent stability across the entire spectrum of attacks, bridging the gap between standard robustness and resilience to destructive generative perturbations (further discussed

Table 4: **Quantitative analysis of the generation quality.** Adding *SERUM* has a negligible effect on the quality of the generated images and comparable to prior approaches.

| Method | FID (↓) | | | | CLIP Score (↑) | | | |
|---|---|---|---|---|---|---|---|---|
| | SD 1.4 | SD 2.0 | SD 2.1 | Avg | SD 1.4 | SD 2.0 | SD 2.1 | Avg |
| Clean | 17.74 | 17.53 | 18.43 | 17.90 | 0.3123 | 0.3169 | 0.3137 | 0.3143 |
| TrustMark | 17.80 | 17.51 | 18.45 | 17.92 | 0.312 | 0.3162 | 0.3135 | 0.3139 |
| Stable Signature | 17.87 | 17.60 | 18.44 | 17.97 | 0.3069 | 0.3159 | 0.3140 | 0.3123 |
| RingID | 20.17 | 18.73 | 20.19 | 19.70 | 0.3103 | 0.3147 | 0.3125 | 0.3125 |
| GaussMarker | 20.03 | 19.32 | 21.13 | 20.16 | 0.3115 | 0.3157 | 0.3147 | 0.3140 |
| *SERUM* (Ours) | 19.14 | 18.51 | 19.24 | 18.96 | 0.3115 | 0.3164 | 0.3135 | 0.3138 |

in Appendix E). Overall, these results highlight robustness of our *SERUM* also against dedicated attacks.

## 4.4 RADIOACTIVITY

Watermark radioactivity (Sablayrolles et al., 2020) refers to the persistence or transferability of a watermark when watermarked images are used to train or adapt new generative models. High radioactivity means that the watermark signal remains detectable in outputs from models trained on watermarked data. This is a desirable property to trace data provenance through its lifecycle where generated data can be used in training of new generative models (Sander et al., 2024; Meintz et al., 2025; Kerner et al.,

Table 3: **Radioactivity.**

| Model | Adaptation | TPR @ 1%FPR |
|---|---|---|
| SD 1.4 | LoRA | 52.30% |
| SD 1.4 | FFT | 77.12% |
| SANA | FFT | 96.76 % |

2025). We evaluate *SERUM*'s radioactivity by adapting diffusion models on images generated by watermarked SD 2.1 via full fine-tuning (denoted as FFT) and LoRA (Low Rank Adaptation) (Hu et al., 2022). We present the results in Table 3. For the purpose of this experiment, we train SD 1.4 and SANA-0.6B and report the TPR @ 1% FPR for a checkpoint with the lowest evaluation loss in Table 3. For more details see Appendix I. Our results suggest that *SERUM* is highly radioactive for both SD models and SANA, displaying large impact on diverse model families adapted with different methods on watermarked data. To the best of our knowledge, this is the first instance in which watermarked outputs from diffusion models have been demonstrated to be radioactive.

## 4.5 QUALITY OF WATERMARKED IMAGES

Table 4 reports the impact of watermarking on generation quality, measured by FID and CLIP Score. Our method demonstrates a negligible impact on visual quality across all evaluated Stable Diffusion versions. On average, *SERUM* incurs only a minor increase in FID (17.90 Clean vs. 18.96 Ours) while preserving semantic consistency, as indicated by the nearly unchanged Average CLIP Score (0.3143 Clean vs. 0.3138 Ours).

Regarding baselines, TrustMark and Stable Signature achieve the best fidelity retention, with averages of 17.92 and 17.97 FID, respectively. However, *SERUM* demonstrates superior quality compared to the other distribution-based methods, RingID and GaussMarker, which suffer from higher degradation (19.70 and 20.16 Avg FID, respectively). Overall, these findings highlight that the robustness of *SERUM* does not come at the expense of visual generation quality, as qualitatively shown in Figure 2.

## 4.6 MULTI-USER WATERMARKING

We illustrate the performance of our method in a multi-user setting in Figures 3a and 3b. We train $m = 135$ detectors and evaluate the system with a subset size of $k = 2$, supporting a total user base of $n = \binom{135}{2} = 9,045$. Each pattern uses $\alpha = 0.3$, producing a combined watermark strength of $\alpha_{total} = 0.6$. Evaluation is performed on 25,000 images for User Accuracy and 5,000 images for TPR calculation.

**Watermark detection (TPR @ 1% FPR).** *SERUM* proves highly robust at scale. With 9,045 users, it maintains 99.96% TPR on clean images, 99.76% on Gaussian Blur, and 99.30% on JPEG Com-

| A surfer standing in the ocean holding a surfboard | A dog wearing a bandanna, sitting on the trunk of a car | Two people on a long rowboat in a river or lake | A blue and yellow train, a building, and some cars | A wooden bench sitting in the grass, under a tree | Several sheep are eating hay near a fence |

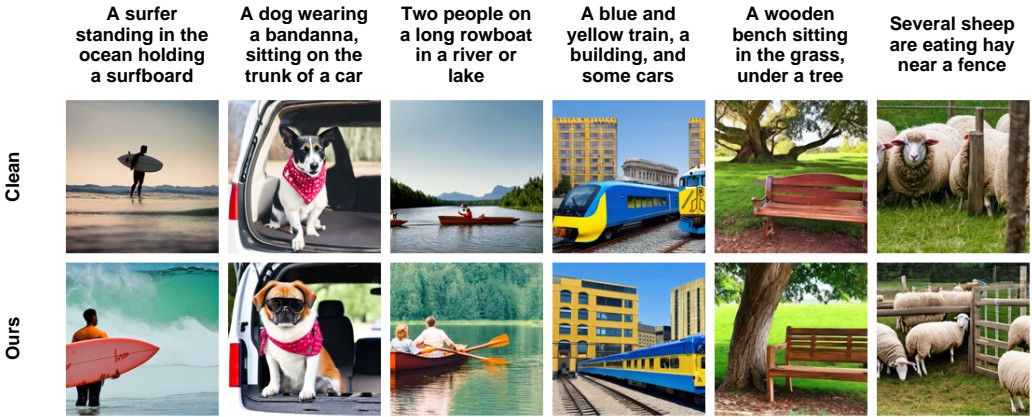

Figure 2: **Qualitative analysis of generation quality.** We present outputs from the SD 2.1 model without (Clean) and with our *SERUM* (denoted as Ours with the parameter $\alpha = 0.5$) watermark. The most important image qualities like style and content are preserved while slightly modifying shape or perspective. More examples can be found in Appendix Figure 8.

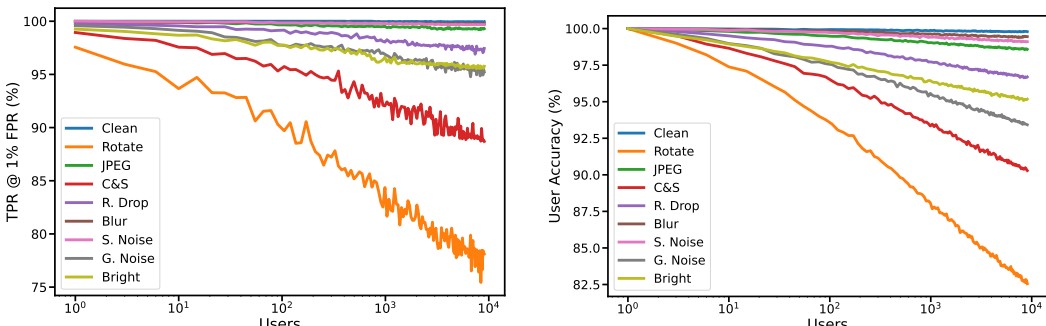

(a) **Watermark detection (TPR @ 1% FPR).** *SERUM* scales robustly to 9,045 users, maintaining over 95% TPR on most perturbations.

(b) **User identification accuracy.** Identification remains precise at scale, achieving over 90% accuracy on 7 of 8 perturbations for 9,045 users.

Figure 3: **Performance of *SERUM* in a multi-user setting.**

pression. Even under challenging geometric perturbations, it retains strong detectability, achieving 88.70% TPR on Crop & Scale and 78.12% on Random Rotation.

**User Identification Accuracy.** *SERUM* consistently maps the signal to the correct user. At the full scale of 9,045 users, the system maintains 99.79% accuracy on clean images and 99.10% on Salt & Pepper Noise. Even under the most difficult transformations, such as Random Rotation and Crop & Scale, the system correctly identifies the user in 82.56% and 90.29% of cases, respectively.

These results highlight the practical utility of *SERUM*: it scales efficiently to thousands of users while accurately identifying ownership even under significant image perturbations.

## 4.7 ABLATION STUDIES

To further examine the internal properties of our *SERUM* watermark, we conduct a series of ablation studies. We analyze how the watermark strength $\alpha$ affects both Fréchet Inception Distance (FID) and detection performance. Furthermore, we dissect the contribution of individual training components (specifically the loss terms and the augmentation sampler temperature) to demonstrate the necessity of our dynamic training strategy. The results highlight that $\alpha = 0.5$ strikes the right balance between fidelity and detection, and that combining precomputed and dynamic augmentations is crucial for achieving state-of-the-art robustness.

Table 5: **Impact of watermark strength** $\alpha$ on image quality (FID) and detection robustness (TPR @ 1% FPR) for SD 2.1. We select $\alpha = 0.5$ as the optimal trade-off.

| $\alpha$ | FID | Perturbation Robustness (TPR @ 1% FPR) | | | | | | | | | |
|---|---|---|---|---|---|---|---|---|---|---|---|
| | | Average | Clean | Rotate | JPEG | C&S | R. Drop | Blur | S. Noise | G. Noise | Bright |
| **0.0** | 18.43 | - | - | - | - | - | - | - | - | - | - |
| **0.1** | 18.91 | 78.49 | 98.64 | 23.92 | 94.60 | 41.18 | 90.76 | 97.30 | 94.82 | 76.14 | 89.02 |
| **0.2** | 18.84 | 91.85 | 99.74 | 66.16 | 97.72 | 80.08 | 97.86 | 99.22 | 98.90 | 92.50 | 94.44 |
| **0.3** | 18.95 | 97.73 | 99.96 | 91.98 | 99.74 | 94.84 | 98.68 | 99.74 | 99.78 | 97.64 | 97.24 |
| **0.4** | 19.05 | 98.92 | 99.98 | 95.50 | 99.78 | 98.38 | 99.32 | 99.92 | 99.76 | 98.62 | 98.98 |
| **0.5** | 19.24 | 99.75 | 100.00 | 99.34 | 99.98 | 99.54 | 99.86 | 100.00 | 100.00 | 99.30 | 99.72 |
| **0.6** | 19.96 | 99.82 | 100.00 | 99.60 | 99.94 | 99.58 | 99.86 | 99.96 | 100.00 | 99.70 | 99.74 |
| **0.7** | 20.82 | 99.90 | 100.00 | 99.54 | 99.98 | 99.86 | 99.94 | 99.98 | 100.00 | 99.84 | 99.96 |

Table 6: **Impact of watermark components and sampler temperature** $\tau$ on robustness (TPR @ 1% FPR) for SD 2.1. We compare: **clean** (training solely on clean images), **precompute** (adding precomputed augmentations), and **transform** (our dynamic sampler). The temperature $\tau$ controls the sampler's focus: $\tau = 0$ strictly prioritizes hard examples, while $\tau = \infty$ samples uniformly.

| Setup | Perturbation Robustness (TPR @ 1% FPR) | | | | | | | | | |
|---|---|---|---|---|---|---|---|---|---|---|
| | Average | Clean | Rotate | JPEG | C&S | R. Drop | Blur | S. Noise | G. Noise | Bright |
| **clean** | 78.68 | 100.00 | 14.94 | 99.86 | 59.80 | 77.44 | 100.00 | 98.52 | 66.86 | 88.74 |
| **precompute** | 97.16 | 100.00 | 79.72 | 99.90 | 97.18 | 99.56 | 99.96 | 99.86 | 99.18 | 99.06 |
| **transform** ($\tau = 0$) | 99.48 | 100.00 | 98.06 | 99.70 | 99.24 | 99.82 | 99.98 | 99.92 | 98.86 | 99.30 |
| **transform** ($\tau = 0.1$) | 99.60 | 100.00 | 99.06 | 99.88 | 99.12 | 99.68 | 100.00 | 99.94 | 99.28 | 99.42 |
| **transform** ($\tau = 0.3$) | 99.64 | 100.00 | 99.00 | 99.94 | 99.38 | 99.66 | 99.96 | 99.94 | 99.44 | 99.48 |
| **transform** ($\tau = 0.5$) | 99.70 | 100.00 | 99.30 | 99.86 | 99.48 | 99.86 | 100.00 | 99.94 | 99.50 | 99.36 |
| **transform** ($\tau = 1.0$) | 99.75 | 100.00 | 99.34 | 99.98 | 99.54 | 99.86 | 100.00 | 100.00 | 99.30 | 99.72 |
| **transform** ($\tau = 2.0$) | 99.67 | 100.00 | 98.34 | 99.90 | 99.56 | 99.82 | 100.00 | 99.98 | 99.70 | 99.76 |
| **transform** ($\tau = \infty$) | 99.34 | 100.00 | 95.88 | 99.90 | 99.36 | 99.84 | 99.98 | 99.98 | 99.58 | 99.52 |

We provide additional comprehensive experiments in Appendix G, covering runtime performance, FPR evaluation on real-world data, ablation studies on the augmentation sampler batch size, generalization to different architectures, and robustness analysis under stricter evaluation protocols.

**Impact of Watermark Strength.** We analyze the impact of the watermark strength hyperparameter $\alpha$ on image quality and detection performance. As illustrated in Table 5, there is a distinct trade-off: increasing $\alpha$ strengthens the watermark signal, which significantly improves the TPR, but comes at the cost of a higher FID score, indicating a slight reduction in the diversity of generated images. We select $\alpha = 0.5$ as the optimal balance between fidelity and robustness for our main evaluations. However, this parameter is flexible; larger values can be employed for applications where detection robustness is prioritized over strict fidelity retention.

**Impact of Individual Components.** We further examine how different training components contribute to robustness. Specifically, we evaluate the effect of loss terms on clean, precomputed, and transformed samples, alongside the impact of the augmentation sampler's temperature. Results in Table 6 demonstrate that incorporating all loss terms is essential for high performance and that setting the temperature to $\tau = 1$ yields the best results. We provide a detailed analysis of these sampling dynamics in Appendix G.2.

## 5 CONCLUSIONS

We introduced *SERUM*, a novel watermarking method for DMs that unifies the strengths of prior approaches into a simple, efficient, and robust framework. By combining watermark noise insertion at the initial diffusion stage with the training of a lightweight yet robust watermark detector, *SERUM* achieves state-of-the-art performance in watermark detection while preserving both the distribution and quality of generated images. *SERUM* delivers stronger robustness to image perturbations and watermark removal attacks than previous methods while requiring only minimal training and enabling efficient watermark injection and detection.

## REPRODUCIBILITY STATEMENT

Our own method and core experiments are fully reproducible through the code provided in the supplementary materials. All evaluations are conducted on publicly available models and datasets hosted on `Hugging Face` or `GitHub`, ensuring accessibility for the community. All competing methods are evaluated using their official repositories (versions as of early September 2025) and recommended instructions, with additional training performed when necessary for completeness.

## ACKNOWLEDGMENTS

Adam Dziedzic was supported by the German Research Foundation (DFG) within the framework of the Weave Programme under the project titled "Protecting Creativity: On the Way to Safe Generative Models" with number 545047250. We also acknowledge our sponsors, who support our research with financial and in-kind contributions: OpenAI and G-Research. Additionally, we thank members of the SprintML group for their feedback. Responsibility for the content of this publication lies with the authors

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

# A  AUGMENTATION SAMPLER

Our method involves drawing perturbations, applied to images during training, from a dynamically adjusted probability distribution. The implementation of the sampler makes the training prioritize more difficult transformations improving our method's robustness. We denote the number of augmentation candidates by $n$, the per-augmentation priority vector by $\mathbf{p} \in (0,1)^n$, and the following scalar hyperparameters:

$$\varepsilon, \; \varepsilon_{\text{smooth}}, \; \tau, \; \text{base\_lr\_pos}, \; \text{base\_lr\_neg}, \; \text{boost}, \; \beta.$$

For the purpose of our experiments we set

$\varepsilon = 1e-3, \; \varepsilon_{\text{smooth}} = 1e-3, \; \tau = 1, \; \text{base\_lr\_pos} = 0.2, \; \text{base\_lr\_neg} = 0.05, \; \text{boost} = 3, \; \beta = 1.$

## A.1  SAMPLING

The sampling routine (presented in Algorithm 1) converts priorities into a probability distribution with smoothing and temperature, then samples a single augmentation index.

---

**Algorithm 1:** Sampling from the augmentation distribution

---

**Input:** priority vector $\mathbf{p} \in (0,1)^n$, smoothing parameter $\varepsilon_{\text{smooth}} \geq 0$, temperature $\tau > 0$, integer $k \geq 1$

**Output:** a single index

**Compute softened scores:** $s_i \leftarrow (p_i + \varepsilon_{\text{smooth}})^{1/\tau}$ for all $i$;

**Normalize to probabilities:** $q_i \leftarrow s_i / \sum_{j=1}^{n} s_j$ for all $i$;

Sample a single index $i \sim \text{Categorical}(q_1, \ldots, q_n)$;

**return** $i$;

---

## A.2  UPDATE

The update rule (presented in Algorithm 2) adjusts a chosen augmentation's priority $p_i$ depending on whether it produced a `mistake` or not. The algorithm uses adaptive step sizes that depend on the current priority and a multiplicative `boost` term. Finally, priorities are clipped to stay away from exactly 0 or 1.

---

**Algorithm 2:** Updating a sampled augmentation's priority

---

**Input:** chosen index $i \in \{1, \ldots, n\}$, boolean `mistake`, current priority $p_i$, scalars base\_lr$_{\text{pos}}$, base\_lr$_{\text{neg}}$, boost, $\beta$, $\varepsilon$

**Output:** updated $p_i$

**if** `mistake` = ***true*** **then**

    compute adapt factor: $a \leftarrow (1 - p_i)^{\beta}$;

    learning rate: $\eta \leftarrow \text{base\_lr}_{\text{pos}} \cdot (1 + \text{boost} \cdot a)$;

    increase priority: $p_i \leftarrow p_i + \eta \cdot (1 - p_i)$;

**else**

    compute adapt factor: $a \leftarrow p_i^{\beta}$;

    learning rate: $\eta \leftarrow \text{base\_lr}_{\text{neg}} \cdot (1 + \text{boost} \cdot a)$;

    decrease priority: $p_i \leftarrow p_i - \eta \cdot p_i$;

clip priority: $p_i \leftarrow \text{clip}(p_i, \varepsilon, 1 - \varepsilon)$;

**return** *updated* $p_i$, $t_i^{chosen}$, $t_i^{mistake}$;

---

# B  DERIVATION OF KULLBACK–LEIBLER DIVERGENCE

Let $d = c \cdot w \cdot h$ and let

$$p(x) = \mathcal{N}(x \mid \mathbf{0}, I)$$

be the $d$-dimensional standard Gaussian distribution.

### B.1 KL DIVERGENCE FOR *SERUM*

Let $q_{\text{ours}}(x)$ be the PDF of the initial watermarked noise distribution $\eta'$ for our method.

$$\eta' = \sqrt{1-\alpha}\eta + \sqrt{\alpha}A', \qquad \eta \sim \mathcal{N}(\mathbf{0}, I)$$

Thus $\eta' \sim \mathcal{N}(\sqrt{\alpha}\,A', (1-\alpha)I)$. Using the closed-form KL divergence between Gaussians

$$D_{KL}(q_{\text{ours}} \,\|\, p) = \frac{1}{2}\left[\text{tr}\big(I^{-1}(1-\alpha)I\big) - d + \sqrt{\alpha}\,A'^{\mathsf{T}}I^{-1}\sqrt{\alpha}\,A' + \log\frac{\det I}{\det\big((1-\alpha)I\big)}\right]$$

$$= \frac{1}{2}\left[(1-\alpha)\,d - d + \alpha\|A'\|_2^2 - d\log(1-\alpha)\right]$$

Since $A'$ is normalized such that $\|A'\|^2 = d$, this reduces to:

$$D_{KL}(q_{\text{ours}} \,\|\, p) = -\frac{d}{2}\log(1-\alpha)$$

In particular, for $\alpha = \frac{1}{2}$ this evaluates to:

$$\boxed{D_{KL}(q_{\text{ours}} \,\|\, p) = \frac{d}{2}\log 2}$$

$$D_{KL}(q_{\text{ours}} \,\|\, p) \approx 5{,}678 \text{ for } d = 2^{14}$$

### B.2 KL DIVERGENCE FOR GAUSSMARKER

In this work, we focus on the spatial-domain watermarking variant of GaussMarker, where watermark information is embedded by enforcing specific signs of the initial noise. This restriction naturally corresponds to selecting an orthant in $\mathbb{R}^d$, determined by the watermark signal map.

We define the GaussMarker distribution $q_{\text{GM}}(x)$ as the Gaussian distribution truncated to a fixed orthant $O \subset \mathbb{R}^d$. Since the Gaussian is symmetric, the probability mass of any orthant is exactly $2^{-d}$. To obtain a normalized density, we scale by $2^d$.

$$q_{\text{GM}}(x) = \begin{cases} 2^d\, p(x) & \text{if } x \in O, \\ 0 & \text{otherwise.} \end{cases}$$

$$D_{KL}(q_{\text{GM}} \,\|\, p) = \int_{\mathbb{R}^d} q(x)\,\log\frac{q(x)}{p(x)}\,dx$$

$$= \int_O q(x)\,\log\frac{2^d\,p(x)}{p(x)}\,dx$$

$$= \log 2^d \cdot \int_O q(x)\,dx$$

$$= d\log 2$$

$$\boxed{D_{KL}(q_{\text{GM}} \,\|\, p) = d\log 2}$$

$$D_{KL}(q_{\text{GM}} \,\|\, p) \approx 11{,}357$$

### B.3 KL DIVERGENCE COMPARISON

As such, $D_{KL}$ for our method is lower than that of GaussMarker, providing a theoretical explanation for a lower FID value of our method.

## C ADVANCED ATTACKS DETAILS

**VAE.** We utilize the factorized prior model from Ballé et al. (2018) with a quality level of one. The attack involves compressing and subsequently resampling the image using the VAE's encoder-decoder architecture, effectively regenerating the image content.

**Diffusion-based Attacks Setup.** For all diffusion-based baselines (Regeneration and Rinse), we utilize the $256 \times 256$ unconditional version of Guided Diffusion (Dhariwal & Nichol, 2021). We operate on downscaled images using the DDIM sampler and upscale the model outputs afterwards.

**Regeneration.** In this attack, we perturb the image with noise and then denoise it using the last 12 or 30 steps of a 50-step diffusion schedule, yielding the variants Regen-12 and Regen-30, respectively.

**Rinse.** This method applies the Regeneration process iteratively. Specifically, Rinse-4×8 runs Regeneration with 8 DDIM steps four times, and Rinse-2×25 runs Regeneration with 25 DDIM steps twice.

**CtrlRegen.** Following the official implementation (Liu et al., 2025), we generate images starting from random Gaussian noise using the pretrained checkpoints provided by the authors. Unlike standard regeneration attacks that initialize from the watermarked image, CtrlRegen reconstructs the content solely via semantic and spatial guidance, effectively severing the link to the original watermarked latent code.

**Image-to-Video (I2V).** To evaluate robustness against video generation, we employ Stable Video Diffusion (Blattmann et al., 2023) to generate video sequences of 19 frames conditioned on the watermarked image. We perform watermark detection specifically on the final frame (19th frame) to assess robustness against the temporal accumulation of diffusion and decoding artifacts.

We illustrate these advanced attacks with qualitative examples in Figure 6.

## D PREVENTING BINARITY OF CLASSIFIER OUTPUTS

Training on only precomputed transformations and generated data resulted in poor performance when measured with ROC-AUC after a significant number of epochs. We theorized that at least one of the reasons for this phenomenon is classifier binarization, a situation in which the model begins to learn to output class predictions with near-perfect confidence. This causes the output distribution of the model to be concentrated in two points, which makes TPR @ FPR results suboptimal. To address this issue, we added an additional regularizing sub-term to both loss terms. This sub-term ensures that the model outputs correct predictions for randomly transformed training samples. Since this perturbation is random at each step, the classifier is unable to memorize the outputs for these samples, making the model's confidence low for difficult data.

## E *SERUM*'S GENERALIZATION CAPABILITIES

In this section, we investigate the underlying factors contributing to *SERUM*'s strong generalization to unseen attacks by analyzing the impact of training augmentations, diffusion inversion, and the detection space.

Firstly, we check whether the augmentations used for classifier training improve its robustness to unseen attacks. As such, we train *SERUM* after removing the perturbation-related sub-terms from the loss formulation. While we do notice a significant decrease in performance when evaluated on advanced attacks, the results remain high. This suggests that our method's robustness to such attacks is a core feature of the method itself, rather than a consequence of the transformations used during training. We report these results in Table 7.

Table 7: **Robustness to advanced attacks.** TPR @ 1% FPR for a watermark detector trained without any perturbations.

| VAE | Regen-12 | Rinse-4x8 | Regen-30 | Rinse-2x25 | CtrlRegen | I2V |
|-----|----------|-----------|----------|------------|-----------|-----|
| 98.64 | 90.62 | 87.22 | 52.28 | 51.90 | 86.90 | 47.10 |

We then verify the impact of DDIM inversion on latents by calculating the Euclidean distance between latents of the original image and those of the same image perturbed with a random transformation.

Table 8: **Watermark robustness.** Results showing TPR @ 1% FPR on each perturbation for a watermark detector trained without this specific transformation. Evaluated on 5,000 watermarked and clean samples.

| Rotate | JPEG | C&S | R. Drop | Blur | S. Noise | G. Noise | Bright |
|--------|-------|-------|---------|-------|----------|----------|--------|
| 18.46 | 99.96 | 56.34 | 64.02 | 99.96 | 99.88 | 91.70 | 78.02 |

We compare this distance to the Euclidean distance between the corresponding initial noises after performing DDIM inversion. Running this experiment on 1,000 synthetically generated images with SD 2.1 shows that the Euclidean distance increases from 103.18 to 117.47 after inversion. This indicates that DDIM inversion amplifies the effect of augmentations on latents, implying that inversion-free methods possess an inherent advantage.

Lastly, we observe that most perturbations and attacks primarily operate in pixel space and do not modify the semantics of the image. For instance, noise perturbation distorts the image but does not change its high-level semantics. This suggests that latent-based watermarks should be naturally resilient to pixel-space attacks and transformations, giving them a massive advantage.

These deliberations lead us to attribute our method's success against unseen attacks to two main factors:

1. *SERUM*, unlike most other evaluated methods, does not rely on DDIM inversion.
2. The watermark detector operates in latent space rather than pixel space, providing greater robustness than Stable Signature's watermark extractor.

## F  FLEXIBILITY OF *SERUM*

Let $\mathcal{T}_i$ denote the $i$-th perturbation and $\mathcal{T}$ the full set of augmentations used during training. To measure how training on transformations affects robustness to an individual transformation $\mathcal{T}_i$, we train the detector on $\mathcal{T} \setminus \{\mathcal{T}_i\}$ and then report TPR at 1% FPR measured on $\mathcal{T}_i$. Results are shown in Table 8. We observe substantial drops in performance for rotations, C&S, and random drop when those transformations are omitted from training; in particular, rotation is more damaging than many of the other advanced attacks. The fact that our method can learn near-perfect robustness to rotation when it is included in training demonstrates high flexibility; the detector can learn to recover watermarks even under very challenging perturbations.

## G  ADDITIONAL EXPERIMENTAL RESULTS

### G.1  RUNTIME

We evaluate the computational efficiency of *SERUM* against state-of-the-art baselines on an NVIDIA A100 GPU. The results, summarized in Table 9, demonstrate that *SERUM* achieves a superior trade-off between training effort and inference-time latency.

Unlike inversion-based approaches such as RingID and GaussMarker, which suffer from prohibitive detection costs ($>$110 min) due to the necessity of reversing the diffusion process (DDIM inversion), *SERUM* enables rapid detection (2.5 min per 5,000 images). Additionally, *SERUM* achieves near-instantaneous injection (17ms), significantly outperforming baselines that rely on computationally costlier spectral domain operations (FFT) ($\sim$2s). Furthermore, our training phase is significantly more efficient than other learning-based methods, requiring only 9.48 hours compared to 57.5 hours for Stable Signature and 336 hours for TrustMark. This efficiency stems from our lightweight detector design.

### G.2  RESULTS ON REAL DATA

While training on synthetic data ensures fair evaluation, it does not guarantee low false positive rates when applied to real-world images. To assess generalization, we evaluate our model on 5,000

Table 9: **Runtime Performance.** Measured on an NVIDIA A100 GPU. Injection and detection times are reported for a batch of 5,000 images. Training time for *SERUM* includes the overhead for data generation and pre-computing augmentations.

| Method | Training Time | Watermark Injection | Watermark Detection |
|---|---|---|---|
| TrustMark | 336 h | 76 s | 1.34 min |
| Stable Signature | 57.52 h | 0 ms | 1.1 min |
| RingID | - | 1922 ms | 140.7 min |
| GaussMarker | 0.93 h | 2022 ms | 117.0 min |
| *SERUM* (Ours) | 9.48 h | 17 ms | 2.5 min |

Table 10: **Impact of architectures on watermark robustness.** Results are calculated on 10,000 samples (5,000 watermarked and non-watermarked) and TPR @ 1% FPR thresholds are calculated separately for each perturbation. We use SD 3.0 (medium) underpinned by the Diffusion Transformer (DiT) and compare it against the UNet-based SD 2.1.

| Architecture | Model | Perturbation Robustness (TPR @ 1% FPR) | | | | | | | | |
|---|---|---|---|---|---|---|---|---|---|---|
| | | Average | Clean | Rotate | JPEG | C&S | R. Drop | Blur | S. Noise | G. Noise | Bright |
| Transformer | SD 3.0 | 95.24 | 99.82 | 81.18 | 99.50 | 88.64 | 98.66 | 99.60 | 99.10 | 91.86 | 98.76 |
| UNet | SD 2.1 | 99.75 | 100.00 | 99.34 | 99.98 | 99.54 | 99.86 | 100.00 | 100.00 | 99.30 | 99.72 |

randomly sampled images from the LAION-2B and COCO datasets. The model is trained exclusively on synthetic data, and the decision threshold is fixed to the value computed over clean synthetically generated images. Under this setting, the FPR (at 100% TPR on clean synthetic data) is only 0.24% on LAION-2B and 0.00% on COCO, demonstrating that our method transfers effectively to real data.

## G.3    GENERALIZATION TO DIFFUSION TRANSFORMERS

We compare the performance of our *SERUM* watermark for models underpinned by different architectures. Concretely, we show that the method generalizes to the Diffusion Transformer (DiT) architecture which is used in SD 3.0 model and compare its robustness to perturbations against SD 2.1 based on UNet architecture. We think that *SERUM*'s strong performance with a DiT-based image generator can be attributed to the detector's model-specific training. The results are presented in Table 10.

## G.4    ROBUSTNESS TO EXTREME CROP & SCALE

To assess the limits of watermark persistence under severe geometric distortions, we extend our evaluation of the Crop & Scale perturbation to stricter retention rates, ranging from 75% down to 25%. We present the results in Table 11.

We observe a sharp divergence in performance as the attack severity increases. While Stable Signature remains effective at moderate cropping levels ($\geq 50\%$), it degrades rapidly under aggressive conditions, collapsing to near-zero detection at 25% retention. RingID exhibits the most severe vulnerability, failing to achieve meaningful detection rates even at the mildest 75% retention setting. Similarly, TrustMark and GaussMarker exhibit early failure modes, dropping to negligible detection rates as soon as retention falls below 50%. In stark contrast, *SERUM* demonstrates exceptional resilience, maintaining a TPR of 93.54% (at 1% FPR) and 81.52% (at 0% FPR) even when only one-quarter of the original image content remains.

## G.5    EVALUATION AT STRICTER FALSE POSITIVE RATES

Table 11: **Ablations on Crop and Scale.** We present a range of retain values for the Crop and Scale perturbation on SD 2.1. We report both TPR @ 1% FPR and TPR @ 0% FPR.

| Method | Metric | 75% | 50% | 45% | 40% | 35% | 30% | 25% |
|---|---|---|---|---|---|---|---|---|
| TrustMark | TPR@1% | **99.98** | 0.18 | 0.04 | 0.06 | 0.00 | 0.02 | 0.04 |
| | TPR@0% | **99.96** | 0.04 | 0.02 | 0.00 | 0.00 | 0.00 | 0.00 |
| Stable Signature | TPR@1% | 99.96 | **99.72** | 87.64 | 57.76 | 37.68 | 5.80 | 0.94 |
| | TPR@0% | 98.24 | 80.98 | 75.66 | 3.10 | 0.88 | 0.00 | 0.00 |
| RingID | TPR@1% | 5.50 | 0.01 | 0.00 | 0.00 | 0.00 | 0.00 | 0.00 |
| | TPR@0% | 0.12 | 0.01 | 0.00 | 0.00 | 0.00 | 0.00 | 0.00 |
| GaussMarker | TPR@1% | 88.72 | 5.32 | 3.72 | 2.68 | 3.10 | 2.30 | 1.82 |
| | TPR@0% | 18.26 | 0.20 | 0.10 | 0.02 | 0.18 | 0.06 | 0.04 |
| *SERUM* | TPR@1% | 99.54 | 98.98 | **99.08** | **98.86** | **98.76** | **97.22** | **93.54** |
| | TPR@0% | 95.22 | **94.12** | **95.94** | **89.76** | **91.00** | **88.56** | **81.52** |

Table 12: **Watermark robustness against perturbations.** Results are calculated on 10,000 samples (5,000 watermarked and non-watermarked) and TPR @ 0% FPR thresholds are calculated separately for each perturbation. Please note that this table is equivalent to Table 1 but with decreased FPR.

| Method | Model | Perturbation Robustness (TPR @ 0% FPR) | | | | | | | | |
|---|---|---|---|---|---|---|---|---|---|---|
| | | Average | Clean | Rotate | JPEG | C&S | R. Drop | Blur | S. Noise | G. Noise | Bright |
| TrustMark | SD 2.1 | 59.33 | **100.00** | 4.42 | 93.18 | **99.96** | 0.02 | **100.00** | 14.78 | 87.16 | 34.46 |
| | SD 2.0 | 48.12 | 93.28 | 1.98 | 52.18 | 96.0 | 0.00 | 93.02 | 21.0 | 54.18 | 21.44 |
| | SD 1.4 | 59.23 | **100.00** | 3.42 | 93.38 | **100.00** | 0.76 | **100.00** | 15.74 | 85.24 | 34.56 |
| Stable Signature | SD 2.1 | 77.37 | 99.78 | 81.04 | 44.54 | 98.24 | 96.22 | 95.10 | 77.62 | 17.70 | 86.08 |
| | SD 2.0 | 80.80 | 99.98 | 41.44 | 98.72 | **99.88** | 96.34 | 97.66 | 99.90 | 67.76 | 25.54 |
| | SD 1.4 | 81.87 | 99.92 | 88.14 | 38.96 | 99.54 | 99.52 | 95.12 | 91.34 | 35.70 | 88.68 |
| RingID | SD 2.1 | 85.88 | 100.00 | 98.14 | 99.88 | 0.12 | 99.92 | 99.98 | **99.86** | 79.00 | 95.98 |
| | SD 2.0 | 85.48 | 100.00 | 99.46 | 99.90 | 0.48 | 99.80 | 100.00 | **99.98** | 74.44 | 95.24 |
| | SD 1.4 | 85.48 | 100.00 | 99.58 | 99.85 | 0.02 | 99.66 | 100.00 | **99.75** | 77.59 | 92.87 |
| GaussMarker | SD 2.1 | 23.59 | 12.12 | 12.64 | 11.38 | 18.26 | **100.00** | 3.06 | 16.78 | 12.44 | 37.70 |
| | SD 2.0 | 31.45 | 63.42 | 32.08 | 2.10 | 1.16 | **99.98** | 2.08 | 66.76 | 1.01 | 14.56 |
| | SD 1.4 | 26.54 | 44.38 | 3.24 | 17.36 | 4.62 | **99.98** | 8.48 | 38.78 | 0.90 | 21.16 |
| *SERUM* (Ours) | SD 2.1 | **96.34** | 99.96 | 89.38 | 99.56 | 95.22 | 97.12 | 99.78 | 99.76 | **89.96** | **96.28** |
| | SD 2.0 | **97.19** | 100.00 | 90.16 | 99.70 | 94.32 | 98.18 | 99.76 | 99.46 | **96.26** | **96.88** |
| | SD 1.4 | **96.36** | 100.00 | 90.12 | 99.10 | 92.84 | 98.68 | 99.90 | 99.48 | **90.92** | 96.18 |

To distinguish performance differences masked at the standard 1% FPR, we evaluate robustness at the stricter 0% FPR threshold in Table 12. This analysis reveals substantial fragility in baseline methods: GaussMarker's performance collapses (Average TPR drops to 23.59% TPR on SD 2.1), while RingID exhibits a critical vulnerability to Crop & Scale (0.12% TPR), enabling attackers to easily bypass the watermark. In contrast, *SERUM* demonstrates superior stability, maintaining an average TPR over 96% across all models and avoiding the catastrophic failures observed in prior approaches.

## G.6  IMPACT OF AUGMENTATION SAMPLER BATCH SIZE.

We analyze the effect of the augmentation sampler batch size $m$, which determines the number of dynamic perturbations generated per training step. As detailed in Table 13, increasing $m$ yields consistent improvements in robustness, particularly for challenging geometric transformations. For instance, increasing $m$ from 0 (precomputed only) to 4 boosts robustness against Rotation from 79.72% to 99.34%. However, this improvement comes with a computational overhead. We observe diminishing returns beyond $m = 4$; while scaling to $m = 32$ marginally increases average TPR

Table 13: **Ablation of the augmentation sampler batch size** ($m$)**.** We report TPR @ 1% FPR for the SD 2.1 model, evaluated on 10,000 samples. The precompute row denotes the baseline where only clean and precomputed augmentations are applied (equivalent to $m = 0$). Increasing $m$ consistently improves robustness at the cost of training time.

| Batch Size | Time | Perturbation Robustness (TPR @ 1% FPR) | | | | | | | | | |
|---|---|---|---|---|---|---|---|---|---|---|---|
| | | Average | Clean | Rotate | JPEG | C&S | R. Drop | Blur | S. Noise | G. Noise | Bright |
| precompute | 2.63 h | 97.16 | 100.00 | 79.72 | 99.90 | 97.18 | 99.56 | 99.96 | 99.86 | 99.18 | 99.06 |
| $m = 2$ | 5.66 h | 99.52 | 100.00 | 99.22 | 99.84 | 99.00 | 99.60 | 99.96 | 99.92 | 99.08 | 99.04 |
| $m = 4$ | 9.48 h | 99.75 | 100.00 | 99.34 | 99.98 | 99.54 | 99.86 | 100.00 | 100.00 | 99.30 | 99.72 |
| $m = 8$ | 15.69 h | 99.80 | 100.00 | 99.74 | 99.90 | 99.62 | 99.80 | 100.00 | 99.94 | 99.50 | 99.74 |
| $m = 16$ | 28.15 h | 99.83 | 100.00 | 99.76 | 99.88 | 99.70 | 99.92 | 99.96 | 99.98 | 99.60 | 99.70 |
| $m = 32$ | 59.35 h | 99.82 | 100.00 | 99.42 | 99.96 | 99.68 | 99.96 | 100.00 | 99.96 | 99.64 | 99.80 |

to 99.82%, it inflates training time by over 6 times (from 9.5h to 60h). Consequently, we adopt $m = 4$ as the optimal operating point, providing a favorable balance between training efficiency and state-of-the-art robustness.

### G.7 ANALYSIS OF AUGMENTATION SAMPLER TEMPERATURE

We analyze the impact of the augmentation sampler's temperature $\tau$ on detection robustness, with quantitative results detailed in Table 6. This parameter governs the sharpness of the sampling distribution, controlling the critical balance between efficient hard-negative mining and optimization stability.

At low temperatures ($\tau \to 0$), the sampler becomes overly greedy, almost exclusively selecting the perturbation with the highest loss. This introduces optimization instability, as the model sequentially overfits to the current worst-case distortion instead of learning a generalized representation. Consequently, this lack of batch diversity leads to high-variance updates and causes the model to forget previously mastered transformations.

Conversely, as $\tau \to \infty$, the strategy approaches uniform sampling, which proves inefficient. The model expends significant training capacity on easy augmentations that are already mastered and provide negligible learning signal. This prevents the model from focusing sufficiently on the difficult perturbations required to refine the decision boundary.

Empirically, we find that $\tau = 1.0$ strikes the optimal balance. This setting biases the detector toward high-loss samples to maximize gradient utility, while retaining enough batch diversity to prevent overfitting to specific noise patterns.

## H ADDITIONAL INFORMATION ABOUT GAUSSMARKER

### H.1 EVALUATION DETAILS

To ensure a fair evaluation of GaussMarker, we modified its training procedure to incorporate our adjusted version of the C&S augmentation. In the original procedure, crops were always assumed to be centered in the middle of the image. This is a highly optimistic assumption, as an attacker may choose to place a crop elsewhere. To account for this, we evaluate all methods using a version of this transformation where crops occur at random locations.

For other perturbations that differ from those in the original GaussMarker paper, higher Gaussian Noise and Gaussian Blur instead of the Median Filter; we do not include them in training. These augmentations change the latent's variance, invalidating GaussMarker's approximation used for GNR training. For the same reason, the original authors also did not train the GNR on both the Gaussian Noise and the Median Filter.

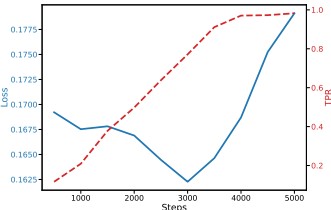 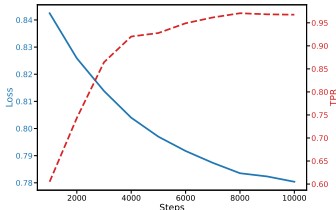 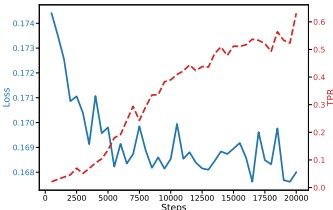

(a) **Stable Diffusion 1.4 FFT.** Checkpoint with minimum evaluation loss (3,000 steps) is used for reporting TPR.

(b) **SANA-0.6B.** Loss decreases steadily across epochs; final checkpoint is used for reporting TPR.

(c) **Stable Diffusion 1.4 LoRA.** Checkpoint with minimum evaluation loss (19,500 steps) is used for reporting TPR.

Figure 4: **Evaluation of radioactivity.**

## H.2 IMPACT OF SAMPLE SIZE ON EVALUATION

The authors of *GaussMarker* report their Fréchet Inception Distance (FID) using only 5,000 samples, whereas at least 10,000 samples are typically recommended for a statistically reliable estimate. A more concerning issue arises in their evaluation of TPR at 1% FPR, as this metric was computed on merely 1,000 samples. In this setting, the threshold at 1% FPR is determined by only 10 negative examples, which we argue is far too few to yield a reliable estimate. In contrast, we performed this evaluation on 5,000 samples and found that the resulting performance is slightly worse than what the original authors report. This suggests that their results may be overly optimistic due to the insufficient sample size, which may also explain the discrepancies between their reported results and ours.

## I RADIOACTIVITY

### I.1 EVALUATION SETUP

For both models we perform full fine-tuning (without the text-encoder) and for SD 1.4 we also adapt it with LoRA on 5,000 training samples, generated by watermarked Stable Diffusion 2.1 with $\alpha = \frac{1}{2}$. For every training checkpoint, we report TPR @ 1% FPR and evaluation loss. Specifically, TPR is evaluated on 5,000 images, while evaluation loss is computed as the MSE of the denoising prediction at random diffusion timesteps on 1,000 images.

We believe that evaluating loss on unseen images is essential to claim radioactivity. In particular, a model that has memorized a watermarked image would yield artificially perfect radioactivity results. For this reason, we expect that most engineers would stop training before the model enters this stage. We therefore report the TPR at the checkpoint with the lowest evaluation loss, which we consider a more realistic estimate.

For TPR @ 1% FPR we utilize a threshold calculated over clean images. We choose this particular metric to have a direct comparison to images generated with a real watermarked model.

For full fine-tuning **Stable Diffusion**, we chose to use a learning rate equal to 5e-6. For **LoRA** we utilized rank 4 and learning rate equal to 1e-5. To train **SANA** we set the learning rate to 1.8e-5. All training runs use a batch size of 8.

### I.2 RESULTS FOR STABLE DIFFUSION 1.4 FFT

We present the results of this experiment in Figure 4a. We observe that SD 1.4 achieves 77.12% TPR for the checkpoint with minimum evaluation loss. This result demonstrates that *SERUM* is highly radioactive for this model.

Continued training results in higher loss, suggesting that the model may be beginning to memorize its training samples. Therefore, we believe that TPR values at later checkpoints may not accurately reflect the true radioactivity of the watermark.

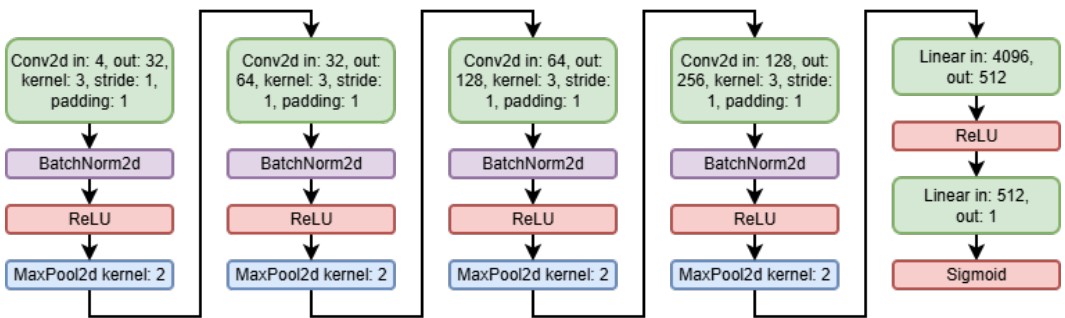

Figure 5: **Watermark detector architecture.**

## I.3 RESULTS FOR SANA-0.6B

For these results, we refer to Figure 4b. In contrast to Stable Diffusion, we do not observe SANA overfitting throughout training. Accordingly, we select the final available checkpoint, which achieves a high TPR of 96.76%. This suggests that *SERUM* is not only radioactive for SD models, but rather for a variety of diffusion models.

## I.4 RESULTS FOR STABLE DIFFUSION 1.4 LoRA

The results for this setting are presented in Figure 4c. At the checkpoint with the lowest evaluation loss, it attains a TPR of 52.30%. This shows that *SERUM* remains distinctly radioactive even when the model is trained using parameter-efficient fine-tuning methods such as LoRA.

## J WATERMARK DETECTOR ARCHITECTURE

The watermark detector is a lightweight Convolutional Neural Network with 2,487,841 parameters. The exact architecture is depicted in Figure 5.

## K LLM USAGE DECLARATION

Large language models were used exclusively to enhance readability, specifically for style, grammar, and spelling, without altering the original meaning of the manuscript.

## L ETHICS STATEMENT

Our work does not raise any direct ethical concerns. On the contrary, *SERUM* is designed as a defensive tool to promote transparency and accountability by enabling reliable identification of images generated by diffusion models. Such capabilities can help mitigate risks associated with the misuse of generative models, including phishing and the spread of misinformation. The method is computationally lightweight, resulting in a negligible carbon footprint.

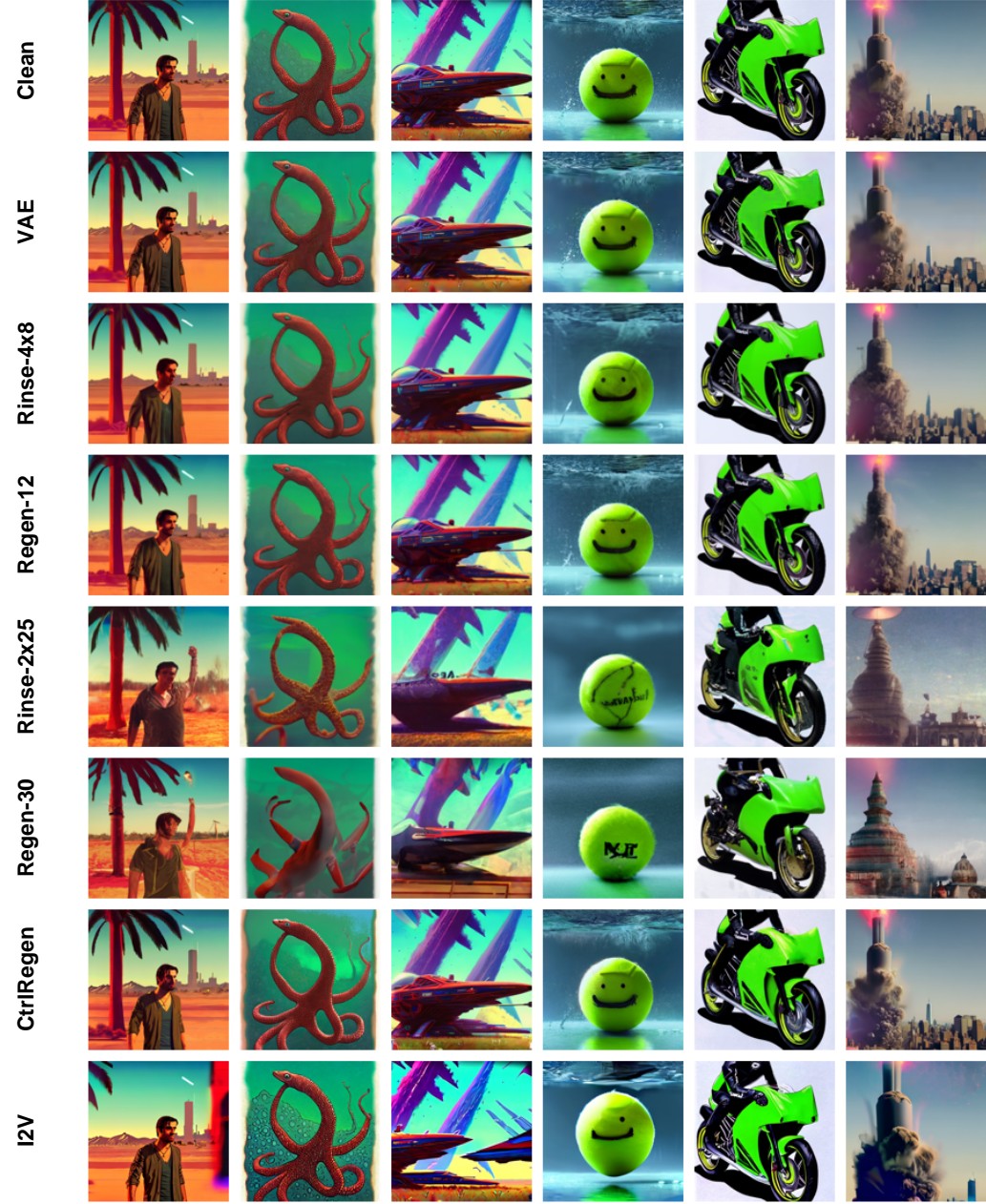

Figure 6: **Visualization of advanced attacks.** Qualitative examples of advanced watermark removal attacks.

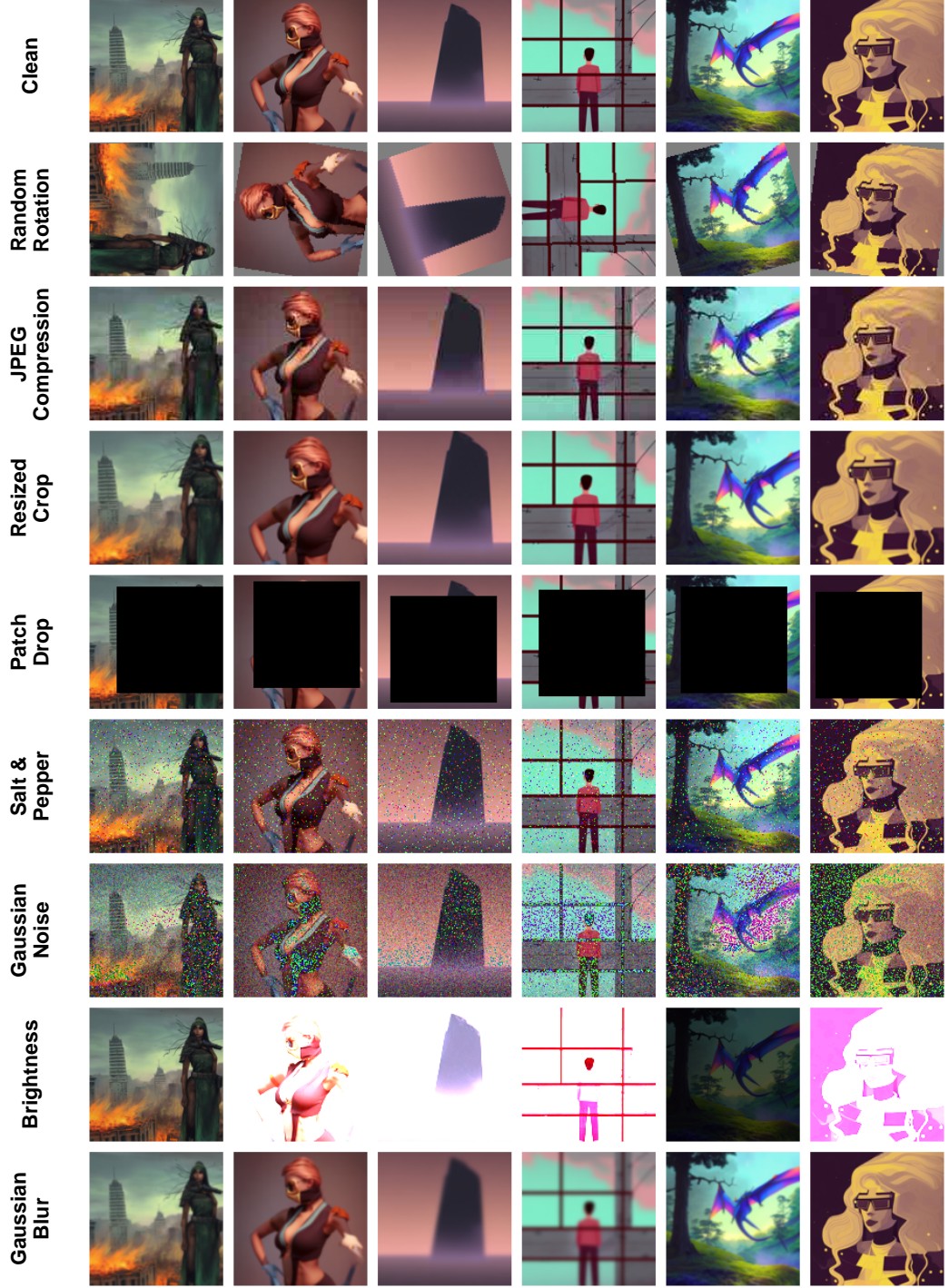

Figure 7: **Visualization of augmentations.** We show qualitative examples of augmentations used to perturb the watermarked and non-watermarked images.

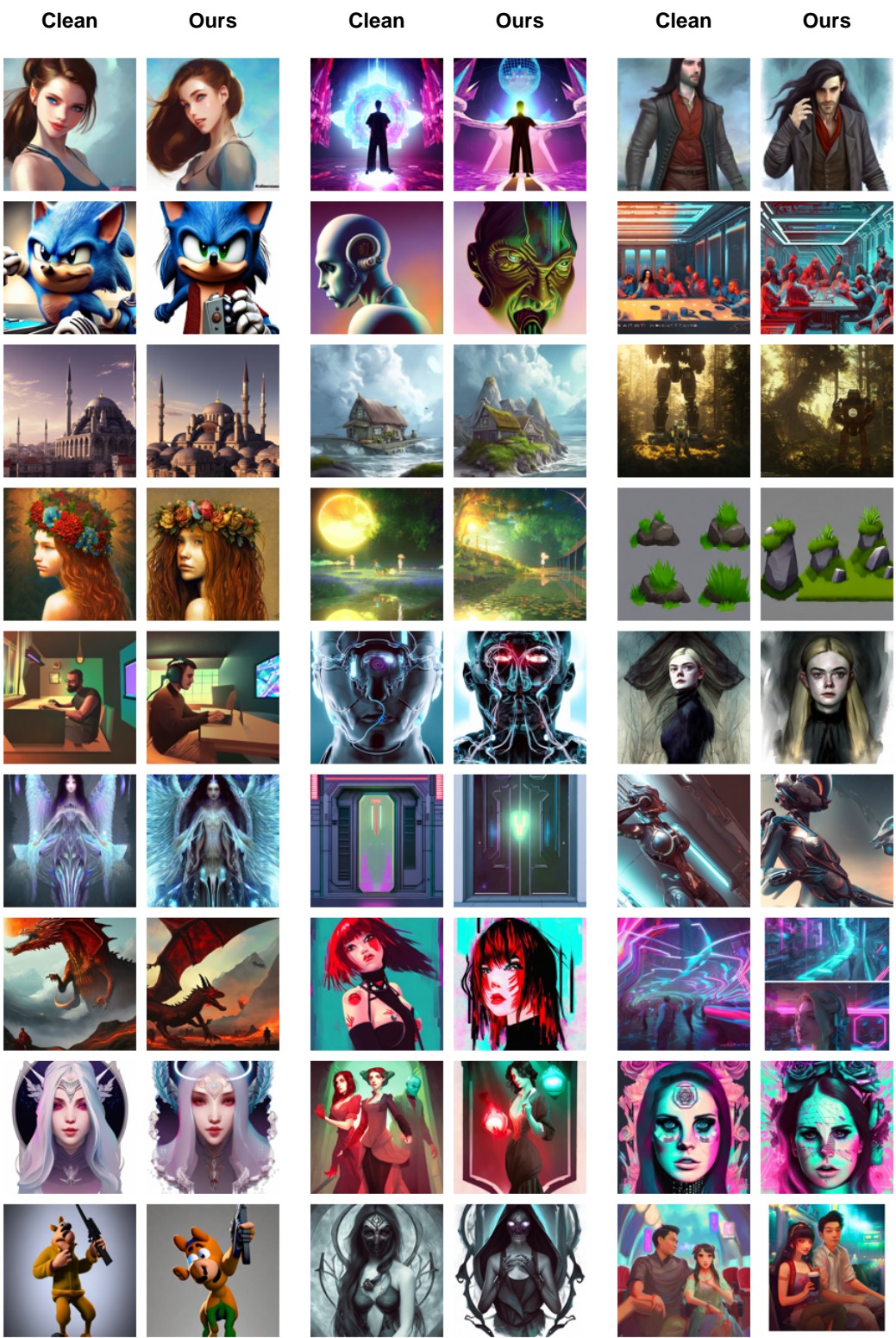

Figure 8: **More qualitative results.** Clean vs Images watermarked with our *SERUM*.

