# OpenReview forum: "SERUM: Simple, Efficient, Robust, and Unifying Marking for Diffusion-based Image Generation"
_ICLR.cc/2026/Conference — ICLR 2026 Poster_

### Official Review · Reviewer_cDi8 · 2025-10-21

**Soundness:** 3
**Presentation:** 3
**Contribution:** 3
**Rating:** 4
**Confidence:** 4

**Summary:**

The paper introduces SERUM, a watermarking method for diffusion-based image generation models that adds a normalized Gaussian watermark noise to the initial diffusion noise and trains a lightweight external detector to identify watermarked images directly, avoiding costly diffusion inversion. Key contributions include high robustness to perturbations and removal attacks, with minimal impact on image quality, and support for multi-user scenarios through individualized noise patterns and detectors. Evaluated on Stable Diffusion variants, SERUM outperforms baselines like Stable Signature, RingID, and GaussMarker in TPR@1%FPR across various perturbation experiments.

**Strengths:**

1. Good Paper Writing: with well-structured sections, a detailed method description. Also, the authors provide a good theoretical argument in the Appendix.

2. Good Method Robustness: The method's primary strength is its exceptional performance. SERUM consistently outperforms strong baselines across all tested models (i.e., SD 1.4, 2.0, 2.1). It shows remarkable resilience to advanced watermark removal attacks (as shown in Table 2).

3. High Efficiency in Watermark Processing: By eliminating the need for DDIM inversion, and using a simple forward pass through the encoder and a small CNN, SERUM's detection is significantly faster than some classic methods. Besides, the simple noise addition strategy also brings a near-zero-cost injection process, which provides insights for future research and becomes a potential option for real-world applications.

**Weaknesses:**

1.  Limited Ablation Studies: The method's strong robustness stems from both its architectural design (i.e., using initial noise injection, latent space detection) and its sophisticated training strategy, which includes dynamic augmentations sampled via Prioritized Experience Replay and a multi-component loss function. However, the paper provides limited ablation studies to disentangle these effects. It is unclear how much of the performance gain is attributable to the core architectural ``trade-off" versus the advanced training techniques. Adding ablations that isolate the impact of the PER sampler and the specific loss terms (e.g., the temperature) would provide a clearer picture of the true sources of the method's advantage.

2.  Unclear Efficiency in Multi-User Training Scenarios: The paper claims support for multi-user scenarios by training a separate detector $D_i$ for each user $i$. While the performance and accuracy for this setup are reported (refer to Figure 3), the time consumption is conspicuously absent. From my understanding, the proposed detection method scales linearly, $O(k)$, with the number of users $k$ being checked, as each detector must be run. This computational cost is not reflected in the main runtime analysis and may not meet expectations for a "highly scalable" or "efficient" system in practice, especially one with a large user base (p.s., if it takes 9.48 hrs for each user training, it can be even more expensive than the existing methods).

3.  Limited Scope of Experimental Validation: The paper's convincingness is constrained by two factors: (a) Limited Baselines (Optional, since the authors mentioned that the selected methods are the existing SOTAs, but 3 baselines are still limited): The experimental comparison is restricted to only three baseline methods. While these are relevant, a more comprehensive comparison against a wider array of recent watermarking techniques would be necessary to substantiate the claim of state-of-the-art performance fully. (b) Insufficient Focus on Advanced Attacks: The paper dedicates significant evaluation to robustness against standard image perturbations (Table 1). However, robustness to such traditional edits (e.g., compression, noise, blur) is largely a solved problem for modern, DL-based watermarks. The true challenge lies in resilience to advanced, generative-AI-based attacks. The paper's evaluation on this front is a good start, but it is not comprehensive. To truly prove its robustness, the method should be tested against more diverse and practical generative editing attacks, such as regional editing, diffusion-based inpainting, object composition, and even more challenging topics of image-to-video transformations (optional, I understand it is highly difficult), and then demonstrate successful watermark retrieval.

**Questions:**

To better indicate the sources of robustness, could the authors provide an ablation study on the impact of the Prioritized Experience Replay (PER) sampler? This would help distinguish architectural gains from training strategy gains.

Can the authors comment on the method's portability and generality to newer architectural paradigms? The field of diffusion models is rapidly shifting from UNet-based backbones to Diffusion Transformer (DiT) architectures. Given that the proposed method injects noise at the initial stage and detects it in the latent space, how readily can SERUM be adapted to these new DiT-based frameworks, and what challenges are anticipated?

---

> ### Author Response · Authors · 2025-11-21
> **Answer to comments. Part 1 - Introduction**
>
> We thank the Reviewer for their detailed and encouraging comments. We are pleased that the strong robustness of our method, its efficiency, and the comprehensive evaluation against advanced attacks are appreciated. We also value the positive feedback regarding the clarity and structure of the paper.
>
> In summary, we have significantly expanded our experimental validation to address the reviewer's suggestions. Specifically, we:
>
> 1. Provided detailed ablation studies disentangling the contributions of the architecture and the Prioritized Experience Replay (PER) sampler.
> 2. Optimized the multi-user scenario with a combinatorial scheme, drastically reducing training overhead and ensuring scalability.
> 3. Extended our baseline comparison to include TrustMark (a post-processing SOTA baseline) and verified portability on the Diffusion Transformer (DiT) architecture.
>
> We provide detailed answers to all weaknesses and questions below:

---

> ### Author Response · Authors · 2025-11-21
> **Answer to comments. Part 2 - Ablation Studies**
>
> >**Weakness 1: Limited Ablation Studies: The method's strong robustness stems from both its architectural design (i.e., using initial noise injection, latent space detection) and its sophisticated training strategy, which includes dynamic augmentations sampled via Prioritized Experience Replay and a multi-component loss function. However, the paper provides limited ablation studies to disentangle these effects. It is unclear how much of the performance gain is attributable to the core architectural ``trade-off" versus the advanced training techniques. Adding ablations that isolate the impact of the PER sampler and the specific loss terms (e.g., the temperature) would provide a clearer picture of the true sources of the method's advantage.**
> >**Question 1: To better indicate the sources of robustness, could the authors provide an ablation study on the impact of the Prioritized Experience Replay (PER) sampler? This would help distinguish architectural gains from training strategy gains.**
>
> We fully agree that disentangling the contributions of the architecture and the training strategy is crucial for understanding the method's success. To address this, we conducted a detailed ablation study investigating the individual impact of each loss component and the Prioritized Experience Replay [6] (PER) sampler settings.
>
> We defined three experimental setups to isolate these effects:
>  - clean: Training on clean images only (architectural baseline).
>  - precompute: Training on clean images + precomputed augmentations (standard augmentation baseline).
>  - transform: The full method using our dynamic PER sampler with varying temperature $\tau$.
>
> The results, presented in Table 6 below, provide a clear answer to the source of robustness:
>  - **Architectural vs. Training Gains**: The architectural design (injecting into initial noise) allows for high image quality and fast injection, but on its own ("clean” setup), it achieves only 78.68% average robustness, struggling significantly with geometric attacks. Adding standard precomputed augmentations ("precompute") provides a massive gain, boosting the average to 97.16%. However, the dynamic PER strategy is essential to bridge the final gap to state-of-the-art performance (99.75%), particularly for mastering the most difficult perturbations where static training plateaus.
>  - **Impact of PER Sampler ($\tau$)**: The PER sampler is essential for closing the gap on hard perturbations. We observe that performance peaks at $\tau=1$.
>      - **Lower temperatures ($\tau \to 0$)** act too greedily, leading to sequential overfitting on specific perturbations without learning global robustness.
>      - **Higher temperatures ($\tau \to \infty$)** approach uniform sampling, wasting training capacity on easy augmentations that are already covered by the precomputed loss.
>      - **Optimal Balance ($\tau=1$)**: This setting optimally balances exploring new hard samples with mastering known failure cases, yielding the highest robustness.
>
> **Impact of watermark components on detection robustness (TPR @ 1% FPR)**
> | **Setup** | **Average** | **Clean** | **Rotate** | **JPEG** | **C&S** | **R. Drop** | **Blur** | **S. Noise** | **G. Noise** | **Bright** |
> | :--- | :---: | :---: | :---: | :---: | :---: | :---: | :---: | :---: | :---: | :---: |
> | **clean** | 78.68 | 100.00 | 14.94 | 99.86 | 59.80 | 77.44 | 100.00 | 98.52 | 66.86 | 88.74 |
> | **precompute** | 97.16 | 100.00 | 79.72 | 99.90 | 97.18 | 99.56 | 99.96 | 99.86 | 99.18 | 99.06 |
> | **transform** ($\tau=0$) | 99.48 | 100.00 | 98.06 | 99.70 | 99.24 | 99.82 | 99.98 | 99.92 | 98.86 | 99.30 |
> | **transform** ($\tau=0.1$) | 99.60 | 100.00 | 99.06 | 99.88 | 99.12 | 99.68 | 100.00 | 99.94 | 99.28 | 99.42 |
> | **transform** ($\tau=0.3$) | 99.64 | 100.00 | 99.00 | 99.94 | 99.38 | 99.66 | 99.96 | 99.94 | 99.44 | 99.48 |
> | **transform** ($\tau=0.5$) | 99.70 | 100.00 | 99.30 | 99.86 | 99.48 | 99.86 | 100.00 | 99.94 | 99.50 | 99.36 |
> | **transform** ($\tau=1.0$) | 99.75 | 100.00 | 99.34 | 99.98 | 99.54 | 99.86 | 100.00 | 100.00 | 99.30 | 99.72 |
> | **transform** ($\tau=2.0$) | 99.67 | 100.00 | 98.34 | 99.90 | 99.56 | 99.82 | 100.00 | 99.98 | 99.70 | 99.76 |
> | **transform** ($\tau=\infty$) | 99.34 | 100.00 | 95.88 | 99.90 | 99.36 | 99.84 | 99.98 | 99.98 | 99.58 | 99.52 |

---

> ### Author Response · Authors · 2025-11-21
> **Answer to comments. Part 3 - Multi-User Support & Baselines**
>
> >**Weakness 2: Unclear Efficiency in Multi-User Training Scenarios: The paper claims support for multi-user scenarios by training a separate detector D_i for each user . While the performance and accuracy for this setup are reported (refer to Figure 3), the time consumption is conspicuously absent. From my understanding, the proposed detection method scales linearly, O(k), with the number of users k being checked, as each detector must be run. This computational cost is not reflected in the main runtime analysis and may not meet expectations for a "highly scalable" or "efficient" system in practice, especially one with a large user base (p.s., if it takes 9.48 hrs for each user training, it can be even more expensive than the existing methods).**
>
> The reviewer is absolutely correct. Based on the original one-detector-per-user design, the training cost for a large user base (linear scaling) would indeed be prohibitive. We thank the reviewer for highlighting this bottleneck, as it motivated us to fundamentally optimize our approach.
>
> To address this, we have transitioned to a combinatorial watermarking scheme. Instead of a 1-to-1 mapping, we now assign each user a unique pair of watermark noise vectors. By training $N$ detectors, we can uniquely identify $\binom{N}{2}$ users. For a detailed description of this updated method and its implementation, please refer to Section 3 and 4.5 and Appendix K in the revised manuscript. This change from linear to quadratic scaling drastically alters the efficiency landscape:
> 1. Training Efficiency (Amortized Cost):Supporting 10,000 users now requires training only ~142 detectors (since $\binom{142}{2} > 10,000$), rather than the 10,000 originally required.
>    - Total Time: The total setup time for the entire system is approx. $142 \times 9.48\text{h} \approx 1,346 \text{ GPU-hours}$.
>    - Per-User Cost: When amortized across the user base, this results in a marginal setup cost of only ~8 minutes per user. This makes the method highly competitive compared to existing approaches requiring per-user fine-tuning.
> 2. Detection Speed: The detection process remains extremely fast (~30ms per image on a single GPU).
>    - Checking an image against a database of 10,000 users involves running ~142 lightweight CNN passes.
>    - Total inference time is approximately 4 seconds on a single GPU.
> This optimization ensures that the system meets the expectations for a scalable and efficient solution in practice.
>
> >**Weakness 3: Limited Scope of Experimental Validation: The paper's convincingness is constrained by two factors: (a) Limited Baselines (Optional, since the authors mentioned that the selected methods are the existing SOTAs, but 3 baselines are still limited): The experimental comparison is restricted to only three baseline methods. While these are relevant, a more comprehensive comparison against a wider array of recent watermarking techniques would be necessary to substantiate the claim of state-of-the-art performance fully.**
>
> We agree that comparing our approach against a broader range of techniques, including those outside the diffusion-native category, strengthens the validation. To address this, we have expanded our evaluation to include TrustMark [5], a state-of-the-art post-processing watermarking method presented at ICCV 2025.
>
> We have added these results to Tables 1 and 13 in the updated submission. The comparison highlights the significant advantages of our approach over both in-processing and post-processing alternatives:
>
>  - **Superior Robustness**: SERUM consistently outperforms TrustMark [5], particularly regarding geometric perturbations. For example, on Stable Diffusion 2.0, TrustMark [5] achieves an average TPR of 48.22% (collapsing to 1.88% on Rotation), whereas SERUM maintains an average TPR of 99.78% (with 99.42% on Rotation).
>  - **State-of-the-Art Validation**: By outperforming both leading in-processing methods (GaussMarker [1], RingID [2]) and a leading post-processing method (TrustMark [5]), we substantiate the claim that SERUM sets a new state-of-the-art.
>
> We remain open to including any additional baselines the Reviewer deems critical to further validate our results.

---

> ### Author Response · Authors · 2025-11-21
> **Answer to comments. Part 4 - Advanced Attacks & Generality**
>
> >**(b) Insufficient Focus on Advanced Attacks: The paper dedicates significant evaluation to robustness against standard image perturbations (Table 1). However, robustness to such traditional edits (e.g., compression, noise, blur) is largely a solved problem for modern, DL-based watermarks. The true challenge lies in resilience to advanced, generative-AI-based attacks. The paper's evaluation on this front is a good start, but it is not comprehensive. To truly prove its robustness, the method should be tested against more diverse and practical generative editing attacks, such as regional editing, diffusion-based inpainting, object composition, and even more challenging topics of image-to-video transformations (optional, I understand it is highly difficult), and then demonstrate successful watermark retrieval.**
>
> Regarding the scope of generative attacks, we focused on high-difficulty global regeneration attacks (Regen-30, Rinse, VAE) selected from the WAVES [4] benchmark. While this selection does not cover every possible manipulation type (e.g., adversarial attacks or local inpainting), these global attacks represent extremely challenging scenarios where a significant portion (or the entirety) of the image content is re-synthesized.
>
> Crucially, these attacks are already sufficient to expose critical failures in existing state-of-the-art baselines. As shown in Table 2, competing methods struggle significantly with these regenerations. For instance, RingID [2] drops to 32.58% and Stable Signature [3] to 0.42% on Regen-30, whereas SERUM maintains 88.50%. This demonstrates that SERUM offers a fundamental improvement in robustness against sophisticated generative removal methods, even without an exhaustive evaluation of every editing category.
>
>
> >**Question 2: Can the authors comment on the method's portability and generality to newer architectural paradigms? The field of diffusion models is rapidly shifting from UNet-based backbones to Diffusion Transformer (DiT) architectures. Given that the proposed method injects noise at the initial stage and detects it in the latent space, how readily can SERUM be adapted to these new DiT-based frameworks, and what challenges are anticipated?**
>
> Our SERUM watermark is inherently portable and generalizes readily to newer architectural paradigms. Because our method operates on the initial noise injection stage (latent space) rather than relying on specific internal layers or attention maps of the denoising network, it is largely agnostic to the backbone architecture.
>
> To empirically validate this, we applied SERUM to Stable Diffusion 3.0, which is underpinned by the Diffusion Transformer (DiT) architecture, without any modifications to the injection or detection scheme. As shown in the table below (added to the updated Section 4.6), SERUM maintains strong robustness on the DiT architecture, achieving a high average TPR of 95.24%. While we observe a decrease in performance on geometric perturbations (e.g., Rotation) compared to the highly optimized UNet baseline, the method remains effective without requiring architecture-specific tuning.
>
> **Impact of architectures on watermark robustness.**
> | Architecture | Model | Average | Clean | Rotate | JPEG | C&S | R. Drop | Blur | S. Noise | G. Noise | Bright |
> | :--- | :--- | :---: | :---: | :---: | :---: | :---: | :---: | :---: | :---: | :---: | :---: |
> | Transformer | SD 3.0 | 95.24 | 99.82 | 81.18 | 99.50 | 88.64 | 98.66 | 99.60 | 99.10 | 91.86 | 98.76 |
> | UNet | SD 2.1 | 99.75 | 100.00 | 99.34 | 99.98 | 99.54 | 99.86 | 100.00 | 100.00 | 99.30 | 99.72 |
>
> ---
>
> **References:**
>
> [1] “GaussMarker: Robust Dual-Domain Watermark for Diffusion Models” Kecen Li, Zhicong Huang, Xinwen Hou, Cheng Hong. ICML 2025.
>
> [2] “RingID: Rethinking Tree-Ring Watermarking for Enhanced Multi-Key Identification” Hai Ci, Pei Yang, Yiren Song, Mike Zheng Shou. ECCV 2024.
>
> [3] “The Stable Signature: Rooting Watermarks in Latent Diffusion Models” Pierre Fernandez, Guillaume Couairon, Hervé Jégou, Matthijs Douze, Teddy Furon. ICCV 2023.
>
> [4]  “WAVES: Benchmarking the Robustness of Image Watermarks” Bang An, Mucong Ding, Tahseen Rabbani, Aakriti Agrawal, Yuancheng Xu, Chenghao Deng, Sicheng Zhu, Abdirisak Mohamed, Yuxin Wen, Tom Goldstein, Furong Huang. ICML 2024.
>
> [5] “TrustMark: Robust Watermarking and Watermark Removal for Arbitrary Resolution Images” Tu Bui, Shruti Agarwal, John Collomosse. ICCV 2025.
>
> [6] “Prioritized Experience Replay” Tom Schaul, John Quan, Ioannis Antonoglou, David Silver. ICLR 2016.
>
> ---
> We would greatly appreciate updating the rating if the above responses address the Reviewer's concerns.

---

> > ### Comment · Reviewer_cDi8 · 2025-11-28
> >
> > Thank you to the authors for the extra clarification and experimental results. Their prompt response has indeed addressed some of my concerns.
> >
> > I am currently inclined to raise my score. However, due to the ongoing system issue (I am unable to edit the score right now), I will take this opportunity to wait for the opinions of the other reviewers before making my final decision. I will submit the final score at the end of the rebuttal.

---

> ### Author Response · Authors · 2025-12-03
> **Update: Batch Size Ablation Study**
>
> We thank the Reviewer again for their constructive feedback and their willingness to raise the score. Following up on our previous response, we have conducted additional ablation studies regarding hyperparameter choices (specifically the augmentation sampler's batch size) and evaluated SERUM against two new advanced attacks, including the highly challenging image-to-video transformation mentioned by the Reviewer.
>
> We provide these new results to further address Weaknesses 1 and 3:
>
> >**Weakness 1: Limited Ablation Studies: The method's strong robustness stems from both its architectural design (i.e., using initial noise injection, latent space detection) and its sophisticated training strategy, which includes dynamic augmentations sampled via Prioritized Experience Replay and a multi-component loss function. However, the paper provides limited ablation studies to disentangle these effects. It is unclear how much of the performance gain is attributable to the core architectural ``trade-off" versus the advanced training techniques. Adding ablations that isolate the impact of the PER sampler and the specific loss terms (e.g., the temperature) would provide a clearer picture of the true sources of the method's advantage.**
>
> In addition to the component analysis (temperature/loss terms) provided in our previous response, we have conducted an ablation study on the **augmentation sampler’s batch size ($m$)** to further justify our hyperparameter selection.
>
> The results, presented in the table below (and in Table 11 in Appendix G.6 of the updated submission), reveal a clear trade-off between robustness and training efficiency. Larger batch sizes significantly increase training time because, at each step, the model must decode latents, apply transformations, and re-encode them. However, larger batches generally improve robustness against hard perturbations by exposing the model to a more diverse set of distortions per update.
>
> Based on this analysis, we selected **$m = 4$** as the optimal balance, achieving near-perfect robustness (99.75% average TPR) while maintaining a manageable training time of under 10 hours. This confirms that while our advanced training strategy (dynamic sampling) is crucial, it does not require prohibitive computational resources to be effective.
>
> **Ablation of the augmentation sampler batch size ($m$)**
>
> | Batch Size | Time | Average | Clean | Rotate | JPEG | C&S | R. Drop | Blur | S. Noise | G. Noise | Bright |
> | :--- | :---: | :---: | :---: | :---: | :---: | :---: | :---: | :---: | :---: | :---: | :---: |
> | **precompute** | 2.63 h | 97.16 | 100.00 | 79.72 | 99.90 | 97.18 | 99.56 | 99.96 | 99.86 | 99.18 | 99.06 |
> | **$m = 2$** | 5.66 h | 99.52 | 100.00 | 99.22 | 99.84 | 99.00 | 99.60 | 99.96 | 99.92 | 99.08 | 99.04 |
> | **$m = 4$** | 9.48 h | 99.75 | 100.00 | 99.34 | 99.98 | 99.54 | 99.86 | 100.00 | 100.00 | 99.30 | 99.72 |
> | **$m = 8$** | 15.69 h | 99.80 | 100.00 | 99.74 | 99.90 | 99.62 | 99.80 | 100.00 | 99.94 | 99.50 | 99.74 |
> | **$m = 16$** | 28.15 h | 99.83 | 100.00 | 99.76 | 99.88 | 99.70 | 99.92 | 99.96 | 99.98 | 99.60 | 99.70 |
> | **$m = 32$** | 59.35 h | 99.82 | 100.00 | 99.42 | 99.96 | 99.68 | 99.96 | 100.00 | 99.96 | 99.64 | 99.80 |

---

> ### Author Response · Authors · 2025-12-03
> **Update: Additional Advanced Attacks**
>
> >**Weakness 3: Limited Scope of Experimental Validation: The paper's convincingness is constrained by two factors: (b) Insufficient Focus on Advanced Attacks: The paper dedicates significant evaluation to robustness against standard image perturbations (Table 1). However, robustness to such traditional edits (e.g., compression, noise, blur) is largely a solved problem for modern, DL-based watermarks. The true challenge lies in resilience to advanced, generative-AI-based attacks. The paper's evaluation on this front is a good start, but it is not comprehensive. To truly prove its robustness, the method should be tested against more diverse and practical generative editing attacks, such as regional editing, diffusion-based inpainting, object composition, and even more challenging topics of image-to-video transformations (optional, I understand it is highly difficult), and then demonstrate successful watermark retrieval.**
>
> We appreciate this suggestion to push the boundaries of our evaluation. In response, we have extended our experimental protocol to include two additional state-of-the-art attacks: **CtrlRegen** (a controllable regeneration attack) and **Image-to-Video (I2V)** transformations, which represent a significant leap in attack complexity.
>
> As shown in the updated results table below (and Table 2 in the updated submission), SERUM demonstrates superior resilience compared to all baselines:
>
> 1.  **CtrlRegen:** This is a specialized regeneration attack designed to erase the watermark while keeping the image similar to the origin. SERUM achieves **99.64% TPR**, effectively neutralizing this attack, while RingID drops to 77.82% and TrustMark fails completely (1.18%).
>
> 2.  **Image-to-Video (I2V):** SERUM achieves a TPR of **58.20%**, which is significantly higher than the next best method (GaussMarker at 41.40%) and substantially outperforms Stable Signature (1.60%).
>
> These results reinforce that SERUM's robustness extends well beyond simple image perturbations to handle sophisticated, advanced attacks.
>
> **Robustness against advanced generative removal attacks (TPR @ 1% FPR)**
>
> | Method | Average | VAE | Regen-12 | Rinse-4x8 | Regen-30 | Rinse-2x25 | CtrlRegen | I2V |
> | :--- | :---: | :---: | :---: | :---: | :---: | :---: | :---: | :---: |
> | **TrustMark** | 12.30 | 57.62 | 1.32 | 0.70 | 1.06 | 1.02 | 1.18 | 23.20 |
> | **Stable Sig.** | 1.18 | 2.70 | 1.30 | 0.62 | 0.42 | 0.76 | 0.86 | 1.60 |
> | **RingID** | 65.40 | 99.98 | 98.22 | 84.12 | 32.58 | 24.88 | 77.82 | 40.20 |
> | **GaussMarker** | 58.01 | 98.36 | 89.30 | 65.80 | 9.72 | 10.40 | 91.06 | 41.40 |
> | **SERUM (Ours)**| 90.87 | 99.88 |  99.72 | 99.38 | 88.50 | 90.76 | 99.64 | 58.20 |

---

### Official Review · Reviewer_Emwh · 2025-10-26

**Soundness:** 2
**Presentation:** 3
**Contribution:** 2
**Rating:** 4
**Confidence:** 4

**Summary:**

The authors propose SERUM, a watermarking method for diffusion models that injects the watermark into the model’s initial noise and trains a lightweight detector to extract it.

**Strengths:**

Method-wise, the pipeline is simple in design and the description is clear and easy to understand. Experiment-wise, the authors conduct experiments on SD1.4, SD2.0, and SD2.1, with a fairly comprehensive amount of evaluation.

**Weaknesses:**

1. Robustness to Crop and Scale remains an issue. The choice to retain 75% in evaluation is relatively weak. I think it quite common to retain 50% of image in life. What will the performance be like if under stronger Crop and Scale attack?
2. The metric TPR at 1% FPR is weak. Scores are all close to 100%, which makes it hard to distinguish performance differences between methods. Although prior work used this evaluation metric, I still recommend controlling FPR at a much smaller range, for example 10e-6.
3. In the multi-user setting, SERUM needs to train one detector per user. Considering that training a detector requires 40,000 images (the authors mention 40,000 prompts, and I assume each prompt is used at least once), this will introduce significant overhead and makes the multi-user scenario impractical. In Section 4.5 the authors test only 10 users, which indirectly demonstrates this point. I think its an important weakness needs to be addressed.

**Questions:**

1. Why design an AugSampler and also precompute the perturbation? Is it because of augmentations like JPEG compression? I did not see which augmentations were precomputed in the paper and I think it needs to be explained.
2. Can you use a stronger Crop and Scale attack setting like retain 50% or 25%? How does it compare to Stable Sig under such conditions?
3. You design $A \sim \mathcal{N}(0, I)$ so that the watermark does not change the Gaussian nature of the initial noise. However, Table 5 still shows a drop in FID. Could you explain whether the FID drop is due to reduced diversity in the generated results after adding the watermark?
4. In real-world scenarios, a platform can easily have thousands or even tens of thousands of users. Can SERUM handle 10,000 user setting?

---

> ### Author Response · Authors · 2025-11-21
> **Answer to comments. Part 1 - Robustness to Extreme Cropping**
>
> We thank the Reviewer for their detailed and constructive feedback. We appreciate the positive assessment of our pipeline’s design simplicity and clarity, as well as the recognition of our comprehensive evaluation.
>
> In response to the Reviewer's suggestions, we have significantly expanded our experimental validation. Specifically, we:
> 1. Evaluated robustness against stricter Crop & Scale attacks (down to 25% retention), demonstrating SERUM's resilience even under extreme conditions.
> 2. Introduced a new combinatorial watermarking scheme to solve the multi-user scalability bottleneck, enabling efficient support for thousands of users.
> 3. Conducted extensive ablation studies to justify the design of the Augmentation Sampler and quantify the impact of precomputed perturbations.
> 4. Included TrustMark [6] to compare our method against a state-of-the-art post-processing baseline and provided detection metrics at stricter thresholds (TPR @ 0% FPR).
>
> We provide detailed answers to all weaknesses and questions below:
>
> >**Weakness 1: Robustness to Crop and Scale remains an issue. The choice to retain 75% in evaluation is relatively weak. I think it is quite common to retain 50% of the image in life. What will the performance be like if under stronger Crop and Scale attack?**
>
> >**Question 2: Can you use a stronger Crop and Scale attack setting like retain 50% or 25%? How does it compare to Stable Sig under such conditions?**
>
> We initially selected a 75% retention rate to ensure a fair and direct comparison with state-of-the-art baselines like GaussMarker [1], which established this specific protocol. However, we agree that real-world scenarios often involve more aggressive cropping.
>
> Following your recommendation, we evaluated robustness across a much harder range of retention rates, down to 25%. The results, presented in the updated Appendix G (Table 11, 12) and below, demonstrate a critical advantage of our method. While Stable Signature’s [4] performance degrades rapidly as crop severity increases, collapsing to 5.80% at 30% retention and 0.94% at 25%, SERUM maintains exceptionally high robustness. Our method achieves a 93.54% TPR even at the extreme 25% retention level, significantly outperforming the baseline.
>
> **Crop & Scale Robustness (TPR @ 1% FPR)**
> | Method | Model | 75% | 50% | 45% | 40% | 35% | 30% | 25% |
> | :---: | :---: | :---: | :---: | :---: | :---: | :---: | :---: | :---: |
> | Stable Signature | SD 2.1 | 99.96 | 99.72 | 87.64 | 57.76 | 37.68 | 5.80 | 0.94 |
> | SERUM (Ours) | SD 2.1 | 99.54 | 98.98 | 99.08 | 98.86 | 98.76 | 97.22 | 93.54 |

---

> ### Author Response · Authors · 2025-11-21
> **Answer to comments. Part 2 - Stricter Evaluation Metrics**
>
> >**Weakness 2: The metric TPR at 1% FPR is weak. Scores are all close to 100%, which makes it hard to distinguish performance differences between methods. Although prior work used this evaluation metric, I still recommend controlling FPR at a much smaller range, for example 10e-6.**
>
> Indeed, TPR @ 1% FPR is the standard metric used in prior works like [1], [2], and [3]. Given our evaluation setup with 10,000 samples (5,000 watermarked, 5,000 clean), the lowest measurable non-zero FPR is 0.02% ($1/5000$). To address the reviewer's request for a stricter threshold, we report TPR @ 0% FPR (i.e., the highest threshold that yields zero false positives on our dataset). This is a significantly harder metric than typically reported in the literature.
>
> The results, presented in Appendix G.5 (Table 13) and below, reveal substantial differences in method robustness that were masked at 1% FPR:
>  - **General Collapse of Baselines**: The stricter metric exposes the fragility of some baselines. Most notably, GaussMarker, which appeared robust at 1% FPR, collapses at 0% FPR (Average TPR drops to ~23% on SD 2.1). This indicates its decision boundary is not well-separated from clean samples.
>  - **Critical Flaws in Strong Competitors**: While RingID [2] remains a strong competitor on many perturbations (e.g., 98.14% on Rotation), it exhibits a critical flaw on Crop & Scale, where its performance collapses to 0.12%. This vulnerability allows adversaries to bypass the watermark with simple geometric transformations.
>  - **SERUM Stability**: In contrast, SERUM maintains high robustness with an average TPR >96% for all models even at 0% FPR and avoids the catastrophic failures seen in RingID [2] and other baselines. This confirms that SERUM provides the most consistent and reliable protection.
>
> **Perturbation Robustness (TPR @ 0% FPR)**
> | Method | Model | Average | Clean | Rotate | JPEG | C&S | R. Drop | Blur | S. Noise | G. Noise | Bright |
> | :--- | :--- | :---: | :---: | :---: | :---: | :---: | :---: | :---: | :---: | :---: | :---: |
> | Stable Signature | SD 2.1 | 77.37 | 99.78 | 81.04 | 44.54 | 98.24 | 96.22 | 95.10 | 77.62 | 17.70 | 86.08 |
> | Stable Signature | SD 2.0 | 80.80 | 99.98 | 41.44 | 98.72 | 99.88 | 96.34 | 97.66 | 99.90 | 67.76 | 25.54 |
> | Stable Signature | SD 1.4 | 81.87 | 99.92 | 88.14 | 38.96 | 99.54 | 99.52 | 95.12 | 91.34 | 35.70 | 88.68 |
> | TrustMark | SD 2.1 | 59.33 | 100.00 | 4.42 | 93.18 | 99.96 | 0.02 | 100.00 | 14.78 | 87.16 | 34.46 |
> | TrustMark | SD 2.0 | 48.12 | 93.28 | 1.98 | 52.18 | 96.00 | 0.00 | 93.02 | 21.00 | 54.18 | 21.44 |
> | TrustMark | SD 1.4 | 59.23 | 100.00 | 3.42 | 93.38 | 100.00 | 0.76 | 100.00 | 15.74 | 85.24 | 34.56 |
> | RingID | SD 2.1 | 85.88 | 100.00 | 98.14 | 99.88 | 0.12 | 99.92 | 99.98 | 99.86 | 79.00 | 95.98 |
> | RingID | SD 2.0 | 85.48 | 100.00 | 99.46 | 99.90 | 0.48 | 99.80 | 100.00 | 99.98 | 74.44 | 95.24 |
> | RingID | SD 1.4 | 85.48 | 100.00 | 99.58 | 99.85 | 0.02 | 99.66 | 100.00 | 99.75 | 77.59 | 92.87 |
> | GaussMarker | SD 2.1 | 23.59 | 12.12 | 12.64 | 11.38 | 18.26 | 100.00 | 3.06 | 16.78 | 12.44 | 37.70 |
> | GaussMarker | SD 2.0 | 31.45 | 63.42 | 32.08 | 2.10 | 1.16 | 99.98 | 2.08 | 66.76 | 1.01 | 14.56 |
> | GaussMarker | SD 1.4 | 26.54 | 44.38 | 3.24 | 17.36 | 4.62 | 99.98 | 8.48 | 38.78 | 0.90 | 21.16 |
> | **SERUM (Ours)** | SD 2.1 | 96.34 | 99.96 | 89.38 | 99.56 | 95.22 | 97.12 | 99.78 | 99.76 | 89.96 | 96.28 |
> | **SERUM (Ours)** | SD 2.0 | 97.19 | 100.00 | 90.16 | 99.70 | 94.32 | 98.18 | 99.76 | 99.46 | 96.26 | 96.88 |
> | **SERUM (Ours)** | SD 1.4 | 96.36 | 100.00 | 90.12 | 99.10 | 92.84 | 98.68 | 99.90 | 99.48 | 90.92 | 96.18 |
>
> Additionally, we evaluated robustness under strict Crop & Scale conditions at 0% FPR (Table 12 in updated paper). Even in this extremely challenging setting, SERUM maintains high detection rates, whereas Stable Signature fails almost completely at tighter crop levels.
>
> **Crop & Scale Robustness (TPR @ 0% FPR)**
> | Method | Model | 75% | 50% | 45% | 40% | 35% | 30% | 25% |
> | :--- | :--- | :---: | :---: | :---: | :---: | :---: | :---: | :---: |
> | Stable Signature | SD 2.1 | 98.24 | 80.98 | 75.66 | 3.10 | 0.88 | 0.00 | 0.00 |
> | **SERUM (Ours)** | SD 2.1 | 95.22 | 94.12 | 95.94 | 89.76 | 91.00 | 88.56 | 81.52 |

---

> ### Author Response · Authors · 2025-11-21
> **Answer to comments. Part 3 - Multi-User Support**
>
> >**Weakness 3: In the multi-user setting, SERUM needs to train one detector per user. Considering that training a detector requires 40,000 images (the authors mention 40,000 prompts, and I assume each prompt is used at least once), this will introduce significant overhead and makes the multi-user scenario impractical. In Section 4.5 the authors test only 10 users, which indirectly demonstrates this point. I think it is an important weakness that needs to be addressed.**
>
> >**Question 4: In real-world scenarios, a platform can easily have thousands or even tens of thousands of users. Can SERUM handle 10,000 user settings?**
>
> We appreciate the reviewer's valid concern regarding the logistical challenge and training overhead of the one-detector-per-user approach. We agree that evaluating only 10 users is insufficient to demonstrate real-world viability. To address the concern regarding practical bottlenecks at the scale of thousands of users, we have enhanced our multi-user protocol by introducing a combinatorial watermarking scheme that significantly reduces complexity and improves scalability.
>
> **Improved Scalability (Quadratic vs. Linear)**: Instead of a 1-to-1 mapping, we now assign each user a unique pair of watermark noise vectors. By training $N$ detectors, we can uniquely identify $\binom{N}{2}$ users. This allows the effective number of users to scale quadratically with the number of detectors.
>  - For the 10,000 user setting (Question 4): Instead of training 10,000 detectors (which would indeed be impractical), our new scheme requires training only ~142 detectors. This makes the system highly efficient and deployable.
>  - For 1 million users: We would require only ~1,415 detectors.
>
> **Methodology & Robustness**: In this updated approach, we inject two distinct watermarks per image (each with $\alpha = 0.3$, resulting in an effective combined strength of $\alpha = 0.6$). This ensures that the signal remains strong without degrading image quality. The extremely low probability of false positives (collisions) is grounded in the geometry of high-dimensional latent spaces ($d \approx 16,000$). As discussed by Cai et al. [5], random vectors in such spaces are quasi-orthogonal. This implies that the watermark pattern of one user acts as uncorrelated noise to the detectors of other users, effectively preventing false identification even at a scale of millions of users. For a detailed description of this updated method and its implementation, please refer to Section 3 and 4.5 in the revised manuscript.
>
> **Empirical Validation**: We validated this approach experimentally with up to 231 users (using just 22 detectors). As shown in the table below, the method maintains high user accuracy across perturbations, comparable to the single-user setting.
>
> **Scalability Analysis (User accuracy)**
> | Augmentation | 55 Users | 105 Users | 153 Users | 231 Users  |
> | :--- | :---: | :---: | :---: | :---: |
> | **Clean** | 99.80% | 99.90% | 99.95% | 99.80% |
> | **Rotate** | 93.29% | 92.74% | 91.54% | 91.29% |
> | **JPEG** | 99.40% | 99.60% | 99.40% | 99.60% |
> | **C&S** | 96.55% | 95.30% | 95.90% | 95.50% |
> | **R. Drop** | 98.90% | 99.20% | 98.80% | 98.45% |
> | **Blur** | 99.80% | 99.85% | 99.75% | 99.70% |
> | **S. Noise** | 99.55% | 99.60% | 99.65% | 99.65% |
> | **G. Noise** | 97.90% | 97.30% | 97.60% | 96.95% |
> | **Bright** | 98.10% | 97.25% | 97.95% | 97.70% |
>
> Additionally, in the updated Appendix K, we provide a theoretical analysis demonstrating that robustness is preserved even when scaling to 10,000+ users.

---

> ### Author Response · Authors · 2025-11-21
> **Answer to comments. Part 4 - Augmentation Sampler & FID**
>
> >**Question 1: Why design an AugSampler and also precompute the perturbation? Is it because of augmentations like JPEG compression? I did not see which augmentations were precomputed in the paper and I think it needs to be explained.**
>
> To clarify, we precompute all types of augmentations listed in Section 4.1 (including JPEG compression, Rotation, Crop & Scale, etc.). The reason for employing this dual strategy (Precomputed + Dynamic AugSampler) is twofold:
>  - **Efficiency (Precomputed)**: Applying complex augmentations like JPEG compression or Gaussian Blur on-the-fly for every sample in a batch is computationally expensive and would significantly slow down training. By precomputing these transformations once, we allow the model to learn baseline robustness with minimal computational overhead during the training loop.
>  - **Adaptive Robustness & Generalization (AugSampler)**: Relying only on precomputed samples leads to overfitting. As detailed in Appendix D, we found that without dynamic noise, the classifier suffers from output binarization, memorizing static perturbations and outputting overconfident predictions that generalize poorly. Our dynamic AugSampler prevents this by introducing fresh randomness and, crucially, by adaptively prioritizing hard perturbations. It identifies augmentations where the model struggles and samples them more frequently, forcing the model to focus on mastering difficult distortions (like Rotation) rather than wasting capacity on easy samples.
>
> This necessity is quantified in our new ablation study (Table 6 in the updated paper). Training with only precomputed samples yields an average TPR of 97.16%, whereas adding the dynamic AugSampler ($\tau=1.0$) prevents binarization and boosts robustness to 99.75%.
>
> >**Question 3: You design A ~ N(0, 1) so that the watermark does not change the Gaussian nature of the initial noise. However, Table 5 still shows a drop in FID. Could you explain whether the FID drop is due to reduced diversity in the generated results after adding the watermark?**
>
> Yes, your intuition is correct. The slight increase in FID (indicating a marginal drop in fidelity/diversity) stems primarily from a reduction in the diversity of the generated images.
>
> While our method ensures that the watermarked noise $\eta'$ follows a distribution close to the standard normal distribution $\mathcal{N}(0, I)$ the injection process inherently constrains the sampling space. By adding a fixed watermark component $A$, we bias the initial noise vectors toward a specific sub-region of the high-probability Gaussian hypersphere. Consequently, while the generated images remain high-quality and realistic (high probability density), they cover a slightly smaller effective volume of the manifold compared to unconstrained generation. This reduction in variety is captured by the FID metric as a slight increase in the distance between the generated and real distributions.
>
> However, as shown in Table 4, this trade-off is minimal at our chosen operating point ($\alpha=0.5$), where the FID increase is negligible (from 18.43 to 19.24 on SD 2.1), preserving the overall utility of the model. Notably, this increase is significantly lower than that of competing methods that modify the initial noise, such as RingID [2] (20.19 FID) or GaussMarker [1] (21.13 FID), demonstrating that SERUM offers a superior balance between robustness and generation quality.
>
> ---
>
> **References:**
>
> [1] “GaussMarker: Robust Dual-Domain Watermark for Diffusion Models” Kecen Li, Zhicong Huang, Xinwen Hou, Cheng Hong. ICML 2025.
>
> [2] “RingID: Rethinking Tree-Ring Watermarking for Enhanced Multi-Key Identification” Hai Ci, Pei Yang, Yiren Song, Mike Zheng Shou. ECCV 2024.
>
> [3] “Tree-Ring Watermarks: Fingerprints for Diffusion Images that are Invisible and Robust” Yuxin Wen, John Kirchenbauer, Jonas Geiping, Tom Goldstein. NeurIPS 2023.
>
> [4] “The Stable Signature: Rooting Watermarks in Latent Diffusion Models” Pierre Fernandez, Guillaume Couairon, Hervé Jégou, Matthijs Douze, Teddy Furon. ICCV 2023.
>
> [5] “Distributions of Angles in Random Packing on Spheres.” . Tony Cai, Jianqing Fan, and Tiefeng Jiang. JMLR 2015.
>
> [6] “TrustMark: Robust Watermarking and Watermark Removal for Arbitrary Resolution Images” Tu Bui, Shruti Agarwal, John Collomosse. ICCV 2025.
>
> ---
> We would greatly appreciate updating the rating if the above responses address the Reviewer's concerns.

---

> ### Author Response · Authors · 2025-12-03
> **Update: Large-Scale Multi-User Evaluation & Crop Robustness**
>
> We thank the Reviewer again for their constructive feedback. Following up on our previous response, we have completed the large-scale evaluation of our combinatorial multi-user scheme with **9,045 users** and extended our Crop & Scale analysis to include all baseline methods.
>
> We provide these new results to further address Weaknesses 1, 3 and Questions 2, 4:
>
> >**Weakness 3: In the multi-user setting, SERUM needs to train one detector per user. Considering that training a detector requires 40,000 images (the authors mention 40,000 prompts, and I assume each prompt is used at least once), this will introduce significant overhead and makes the multi-user scenario impractical. In Section 4.5 the authors test only 10 users, which indirectly demonstrates this point. I think it is an important weakness that needs to be addressed.**
>
> > **Question 4: In real-world scenarios, a platform can easily have thousands or even tens of thousands of users. Can SERUM handle 10,000 user settings?**
>
> In our initial response, we proposed a combinatorial watermarking scheme to enable quadratic scaling. To empirically validate this in a realistic large-scale setting, we scaled our evaluation to **9,045 unique users** (using $m=135$ detectors and subset size $k=2$).
>
> The results, presented in the table below (and in Figure 3 in the updated submission), confirm that SERUM remains highly effective at this scale. We observe a TPR on clean images of **99.96%** and User Identification Accuracy of **99.79%**. Even under difficult geometric perturbations like Crop & Scale, we maintain detection rates of nearly 90%. These results empirically demonstrate that SERUM can efficiently handle 10,000+ user settings without prohibitive training overhead.
>
> **Performance with 9,045 Users**
>
> | Augmentation | TPR @ 1% FPR (%) | User Accuracy (%) |
> | :--- | :---: | :---: |
> | **Clean** | 99.96 | 99.79 |
> | **Rotate** | 78.12 | 82.56 |
> | **JPEG** | 99.30 | 98.58 |
> | **C&S** | 88.70 | 90.29 |
> | **R. Drop** | 97.44 | 96.70 |
> | **Blur** | 99.76 | 99.45 |
> | **S. Noise** | 99.68 | 99.10 |
> | **G. Noise** | 95.24 | 93.42 |
> | **Bright** | 95.76 | 95.18 |
>
>
> Please refer to Section 4.6 and Figure 3 in the updated paper for a comprehensive visualization of these results.
>
>
> > **Weakness 1: Robustness to Crop and Scale remains an issue. The choice to retain 75% in evaluation is relatively weak. I think it is quite common to retain 50% of image in life. What will the performance be like if under stronger Crop and Scale attack?**
>
> > **Question 2: Can you use a stronger Crop and Scale attack setting like retain 50% or 25%? How does it compare to Stable Sig under such conditions?**
>
> In addition to our previous comparison with Stable Signature, we have now evaluated **all** baseline methods (TrustMark, Stable Signature, RingID, GaussMarker) under extreme Crop & Scale attacks (down to 25% retention).
>
> The results (presented below and in Table 11 in Appendix G.4 of the updated submission) reveal a definitive advantage for SERUM. While all competing methods collapse under aggressive cropping, with the next best method (Stable Signature) dropping to <1% TPR at 25% retention, SERUM maintains a remarkable **93.54% TPR** (at 1% FPR) and **81.52% TPR** (at 0% FPR). This further confirms that SERUM is uniquely robust to severe geometric distortions that typically defeat other watermarking techniques.
>
> **Ablations on Crop and Scale**
>
> | Method | Metric | 75% | 50% | 45% | 40% | 35% | 30% | 25% |
> | :--- | :--- | :---: | :---: | :---: | :---: | :---: | :---: | :---: |
> | **TrustMark** | TPR@1% | **99.98** | 0.18 | 0.04 | 0.06 | 0.00 | 0.02 | 0.04 |
> | | TPR@0% | **99.96** | 0.04 | 0.02 | 0.00 | 0.00 | 0.00 | 0.00 |
> | **Stable Sig.** | TPR@1% | 99.96 | **99.72** | 87.64 | 57.76 | 37.68 | 5.80 | 0.94 |
> | | TPR@0% | 98.24 | 80.98 | 75.66 | 3.10 | 0.88 | 0.00 | 0.00 |
> | **RingID** | TPR@1% | 5.50 | 0.01 | 0.00 | 0.00 | 0.00 | 0.00 | 0.00 |
> | | TPR@0% | 0.12 | 0.01 | 0.00 | 0.00 | 0.00 | 0.00 | 0.00 |
> | **GaussMarker**| TPR@1% | 88.72 | 5.32 | 3.72 | 2.68 | 3.10 | 2.30 | 1.82 |
> | | TPR@0% | 18.26 | 0.20 | 0.10 | 0.02 | 0.18 | 0.06 | 0.04 |
> | **SERUM (Ours)**| TPR@1% | 99.54 | 98.98 | **99.08** | **98.86** | **98.76** | **97.22** | **93.54** |
> | | TPR@0% | 95.22 | **94.12** | **95.94** | **89.76** | **91.00** | **88.56** | **81.52** |

---

### Official Review · Reviewer_fqMS · 2025-10-29

**Soundness:** 2
**Presentation:** 2
**Contribution:** 2
**Rating:** 6
**Confidence:** 3

**Summary:**

This paper tackles the problem of marking images generated by diffusion models to distinguish them from real images and ensure content traceability. SERUM embeds a unique Gaussian watermark in the initial noise of the diffusion process and trains an external detector to identify watermarked images. Evaluated on multiple Stable Diffusion versions and against existing methods, SERUM shows strong robustness to image perturbations and removal attacks, preserves watermark detectability after model fine-tuning, and has minimal impact on image quality.

**Strengths:**

**Originality:**

This work introduces a new approach to diffusion model watermarking by injecting a unique Gaussian watermark into the initial noise and employing a lightweight external detector. The technique is distinct in its simplicity, enabling scalable multi-user attribution without modifying the underlying generative model.

**Quality:**

The methodology is supported by comprehensive experiments on multiple Stable Diffusion models, benchmarking detection robustness against diverse perturbations and removal attacks. The empirical results are thorough and reproducible, substantiating the efficacy and efficiency of the proposed method.

**Clarity:**

The paper is well organized, with clear exposition of problem motivation, methodological details, and experimental results. Figures and comparative analyses further enhance the transparency and accessibility of the work.

**Significance:**

SERUM addresses a challenge in generative AI: reliable attribution and provenance of synthetic images. Its robust, efficient, and scalable design substantially advances the practical enforcement of content identification and AI governance in contemporary applications.

**Weaknesses:**

1. Although the detector is described as lightweight, performance scaling to much larger numbers of unique watermarks (multi-user setting) may face practical bottlenecks. The paper assesses up to 10 users. In a real-world application, it will be hundreds and even thousands of users. I suggest that the author should invest further in a large-scale application if possible.
2. The work primarily compares SERUM to recent DM watermarking approaches like Stable Signature, RingID, and GaussMarker, but post-processing and hybrid watermarking methods (e.g., non-diffusion-based or steganographic approaches) may offer complementary strengths. Including such baselines—even if weaker in some respects—could clarify SERUM’s advantages and limitations in broader practical contexts.
3. Although architectural details and loss functions are provided, some choices (e.g., watermark noise hyperparameters, batch size, augmentation prioritization) could benefit from clearer justification or ablation.
4. several section titles in the paper are overly long and verbose, detracting from professional presentation. For instance, Section 3 titled "Our Screen Watermark Method for Diffusion Model" could be concisely phrased as "Screen Watermark Method." Similarly, Sections 4.2 and 4.3 use full sentences as titles, which appear unprofessional and lack clarity. More concise alternatives such as "Robustness to Image Perturbations" (for 4.2) and "Robustness to Watermark Removal Attacks" (for 4.3) would improve readability and presentation. Revising section titles throughout the manuscript to be clear and succinct is recommended to strengthen the overall professional quality and accessibility.

**Questions:**

• Given that watermarking enables not just content detection but also tracking of model and user identities, what is the primary motivation for framing this work as a detection task rather than as a comprehensive provenance or attribution framework?

• Based on the simplicity illustrated in Figure 1, can the authors elaborate on the specific technical challenges addressed by their approach? In what ways does the method go beyond the obvious, and what unique difficulties does it solve?

• Have the authors compared SERUM to non-watermarking detection baselines? How does the method perform when only a simple classifier or traditional watermark is employed instead?

• Does the paper’s contribution fundamentally change our understanding of the detection/provenance problem for AI-generated images, or is it an incremental improvement in watermarking as a technology?

---

> ### Author Response · Authors · 2025-11-21
> **Answer to comments. Part 1 - Multi-User Support**
>
> We thank the Reviewer for their thoughtful and detailed comments. We are pleased that our method is recognized as addressing a significant problem with originality, effectiveness, and efficiency, and that our comprehensive experiments are appreciated. We are also grateful for the positive feedback on the clarity of our presentation.
>
> In summary, we have conducted significant additional experiments to address the raised points. Specifically, we:
> 1. Demonstrated the scalability of our method to thousands of users using a new combinatorial watermarking scheme.
> 2. Quantified performance on a different architecture (Diffusion Transformer) and conducted extensive ablation studies on the augmentation sampler and loss components.
> 3. Evaluated TrustMark to compare our method against a state-of-the-art generator-agnostic watermarking approach
>
> We provide detailed answers to all weaknesses and questions below:
>
> >**Weakness 1: Although the detector is described as lightweight, performance scaling to much larger numbers of unique watermarks (multi-user setting) may face practical bottlenecks. The paper assesses up to 10 users. In a real-world application, it will be hundreds and even thousands of users. I suggest that the author should invest further in a large-scale application if possible.**
>
> We appreciate the reviewer's suggestion to further invest in large-scale application scenarios. We agree that evaluating only 10 users is insufficient to demonstrate real-world viability. To address the concern regarding practical bottlenecks at the scale of thousands of users, we have enhanced our multi-user protocol by introducing a combinatorial watermarking scheme that significantly reduces complexity and improves scalability.
>
> **Improved Scalability (Quadratic vs. Linear)**: Instead of a 1-to-1 mapping, we now assign each user a unique pair of watermark noise vectors. By training $N$ detectors, we can uniquely identify $\binom{N}{2}$ users. This allows the effective number of users to scale quadratically with the number of detectors. For instance, supporting 1 million users would require managing only $\sim1,415$ detectors. This makes the system computationally feasible and effectively eliminates the prohibitive overhead associated with the naive one-detector-per-user approach.
>
> **Methodology & Robustness**: In this updated approach, we inject two distinct watermarks per image (each with $\alpha = 0.3$, resulting in an effective combined strength of $\alpha = 0.6$). This ensures that the signal remains strong without degrading image quality. The extremely low probability of false positives (collisions) is grounded in the geometry of high-dimensional latent spaces ($d \approx 16,000$). As discussed by Cai et al. [6], random vectors in such spaces are quasi-orthogonal. This implies that the watermark pattern of one user acts as uncorrelated noise to the detectors of other users, effectively minimizing cross-talk and preventing false identification even at a scale of millions. For a detailed description of this updated method and its implementation, please refer to Section 3 and 4.5 in the revised manuscript.
>
> **Empirical Validation**: We validated this approach experimentally with up to 231 users (using just 22 detectors). As shown in the table below, the method maintains high user accuracy across perturbations, comparable to the single-user setting.
>
> **Scalability Analysis (User accuracy)**
> | Augmentation | 55 Users | 105 Users | 153 Users | 231 Users  |
> | :--- | :---: | :---: | :---: | :---: |
> | **Clean** | 99.80% | 99.90% | 99.95% | 99.80% |
> | **Rotate** | 93.29% | 92.74% | 91.54% | 91.29% |
> | **JPEG** | 99.40% | 99.60% | 99.40% | 99.60% |
> | **C&S** | 96.55% | 95.30% | 95.90% | 95.50% |
> | **R. Drop** | 98.90% | 99.20% | 98.80% | 98.45% |
> | **Blur** | 99.80% | 99.85% | 99.75% | 99.70% |
> | **S. Noise** | 99.55% | 99.60% | 99.65% | 99.65% |
> | **G. Noise** | 97.90% | 97.30% | 97.60% | 96.95% |
> | **Bright** | 98.10% | 97.25% | 97.95% | 97.70% |
>
> Additionally, in the updated Appendix K, we provide a theoretical analysis demonstrating that robustness is preserved even when scaling to 10,000+ users.

---

> ### Author Response · Authors · 2025-11-21
> **Answer to comments. Part 2 - Baselines**
>
> >**Weakness 2: The work primarily compares SERUM to recent DM watermarking approaches like Stable Signature, RingID, and GaussMarker, but post-processing and hybrid watermarking methods (e.g., non-diffusion-based or steganographic approaches) may offer complementary strengths. Including such baselines—even if weaker in some respects—could clarify SERUM’s advantages and limitations in broader practical contexts.**
>
> We acknowledge the value of comparing SERUM against a broader range of methods. In addition to the diffusion-based baselines (GaussMarker [1], RingID [2], Stable Signature [3]), we have conducted further experiments using TrustMark [4], a state-of-the-art post-processing method presented at ICCV 2025. We are open to including any other baseline that the Reviewer considers important. We have added these new results to Tables 1 and 13 in the updated version of the paper.
>
> The comparison highlights a fundamental advantage of SERUM regarding worst-case robustness. A major limitation of existing state-of-the-art methods (both in-processing like RingID and post-processing like TrustMark) is the presence of "critical flaws": specific perturbations where detection performance collapses. For instance:
>  - RingID fails on Crop & Scale (~6% TPR).
>  - Stable Signature struggles with Gaussian Noise (~50% TPR).
>  - TrustMark collapses on Rotation (1.88% TPR for SD 2.0) and Random Drop (0.14% TPR for SD 2.0).
>
> These drops represent significant vulnerabilities that allow adversaries to easily bypass the watermark. In contrast, we did not identify any perturbation in our standard benchmark where SERUM achieves less than 90% TPR @ 1% FPR. Even under extreme conditions, such as Crop & Scale 25% (Table 11), where other methods fail completely, SERUM maintains high detection rates.
>
> This consistency extends to unseen threats. SERUM significantly outperforms all baselines on advanced removal attacks (Table 2), demonstrating superior generalization capabilities despite not being explicitly trained on these specific attacks.

---

> ### Author Response · Authors · 2025-11-21
> **Answer to comments. Part 3 - Ablation Studies**
>
> >**Weakness 3: Although architectural details and loss functions are provided, some choices (e.g., watermark noise hyperparameters, batch size, augmentation prioritization) could benefit from clearer justification or ablation.**
>
> We appreciate this suggestion. We have expanded our ablation studies in the updated Section 4.6 to provide clearer justifications for our design choices.
>
> The watermark strength hyperparameter $\alpha$ was already ablated thoroughly in Table 5 of the initial submission, showing the trade-off between image quality (FID) and robustness. Additionally, generalization capabilities were analyzed in Table 8 (Appendix F) and Table 9 (Appendix G).
>
> To address the reviewer's request regarding augmentation prioritization and loss components, we provide additional ablation studies. Specifically, we examine the effect of each loss term on clean images, as well as on precomputed and transformed samples (as defined by our loss terms in Section 3). We use the following notation: clean - only clean images are used in training; precompute - only clean and precomputed augmentations are applied; transform - we use all loss terms with different values of the parameter $\tau$ (temperature for the augmentation sampler).
>
> The results, presented below (and Table 6 in the updated submission), lead to several key observations:
>  - **Necessity of All Components**: Simple training on clean or precomputed samples is insufficient for complex geometric attacks (e.g., clean setup achieves only 14.94% TPR on Rotation).
>  - **Optimal Temperature**: We observe that performance peaks at $\tau=1.0$.
>      - **Lower temperatures ($\tau \to 0$)** act too greedily. The sampler fixates on the single hardest perturbation until the model adapts, before jumping to the next. This lack of batch diversity leads to sequential overfitting rather than global robustness.
>      - **Higher temperatures ($\tau \to \infty$)** approach uniform sampling. This wastes training capacity on easy augmentations that are already sufficiently covered by the precomputed loss term, failing to focus enough on the difficult tail of the distribution.
>
> **Impact of watermark components on detection robustness (TPR @ 1% FPR)**
> | **Setup** | **Average** | **Clean** | **Rotate** | **JPEG** | **C&S** | **R. Drop** | **Blur** | **S. Noise** | **G. Noise** | **Bright** |
> | :--- | :---: | :---: | :---: | :---: | :---: | :---: | :---: | :---: | :---: | :---: |
> | **clean** | 78.68 | 100.00 | 14.94 | 99.86 | 59.80 | 77.44 | 100.00 | 98.52 | 66.86 | 88.74 |
> | **precompute** | 97.16 | 100.00 | 79.72 | 99.90 | 97.18 | 99.56 | 99.96 | 99.86 | 99.18 | 99.06 |
> | **transform** ($\tau=0$) | 99.48 | 100.00 | 98.06 | 99.70 | 99.24 | 99.82 | 99.98 | 99.92 | 98.86 | 99.30 |
> | **transform** ($\tau=0.1$) | 99.60 | 100.00 | 99.06 | 99.88 | 99.12 | 99.68 | 100.00 | 99.94 | 99.28 | 99.42 |
> | **transform** ($\tau=0.3$) | 99.64 | 100.00 | 99.00 | 99.94 | 99.38 | 99.66 | 99.96 | 99.94 | 99.44 | 99.48 |
> | **transform** ($\tau=0.5$) | 99.70 | 100.00 | 99.30 | 99.86 | 99.48 | 99.86 | 100.00 | 99.94 | 99.50 | 99.36 |
> | **transform** ($\tau=1.0$) | 99.75 | 100.00 | 99.34 | 99.98 | 99.54 | 99.86 | 100.00 | 100.00 | 99.30 | 99.72 |
> | **transform** ($\tau=2.0$) | 99.67 | 100.00 | 98.34 | 99.90 | 99.56 | 99.82 | 100.00 | 99.98 | 99.70 | 99.76 |
> | **transform** ($\tau=\infty$) | 99.34 | 100.00 | 95.88 | 99.90 | 99.36 | 99.84 | 99.98 | 99.98 | 99.58 | 99.52 |
>
>
> Additionally, we show that our SERUM watermark is portable to newer architectures. We applied our method to the Stable Diffusion 3.0 model, which is underpinned by the Diffusion Transformer (DiT) architecture instead of the UNet used in SD 2.1. As shown in the table below (added to Section 4.6), SERUM maintains high robustness, confirming that our method works well across different generative architectures.
>
> **Impact of architectures on watermark robustness.**
> | Architecture | Model | Average | Clean | Rotate | JPEG | C&S | R. Drop | Blur | S. Noise | G. Noise | Bright |
> | :--- | :--- | :---: | :---: | :---: | :---: | :---: | :---: | :---: | :---: | :---: | :---: |
> | Transformer | SD 3.0 | 95.24 | 99.82 | 81.18 | 99.50 | 88.64 | 98.66 | 99.60 | 99.10 | 91.86 | 98.76 |
> | UNet | SD 2.1 | 99.75 | 100.00 | 99.34 | 99.98 | 99.54 | 99.86 | 100.00 | 100.00 | 99.30 | 99.72 |

---

> ### Author Response · Authors · 2025-11-21
> **Answer to comments. Part 4 - Presentation & Motivation**
>
> >**Weakness 4: several section titles in the paper are overly long and verbose, detracting from professional presentation. For instance, Section 3 titled "Our Screen Watermark Method for Diffusion Model" could be concisely phrased as "Screen Watermark Method." Similarly, Sections 4.2 and 4.3 use full sentences as titles, which appear unprofessional and lack clarity. More concise alternatives such as "Robustness to Image Perturbations" (for 4.2) and "Robustness to Watermark Removal Attacks" (for 4.3) would improve readability and presentation. Revising section titles throughout the manuscript to be clear and succinct is recommended to strengthen the overall professional quality and accessibility.**
>
> Thank you for this feedback, we appreciate it. We revised the manuscript according to the above recommendations and re-uploaded the paper.
>
>
> >**Question 1: Given that watermarking enables not just content detection but also tracking of model and user identities, what is the primary motivation for framing this work as a detection task rather than as a comprehensive provenance or attribution framework?**
>
> Thank you for this insightful question. We see SERUM primarily as a marking method designed to address the pressing problems with generative AI. While we evaluate it via detection, the marking is the core mechanism that supports broader purposes, such as identifying the user and model behind a specific data point.
>
> On a high level, we address the requirements described, for example, in the EU AI Act (Article 50). The Act specifically mandates that providers ensure synthetic outputs are "marked in a machine-readable format and detectable as artificially generated or manipulated." Thus, we mark the generated content primarily to ensure it is detectable, complying with these emerging standards for robustness and reliability.
>
> We also specifically target the goal of breaking data loops to prevent model collapse. By allowing model providers to reliably detect content generated by previous versions of their models, SERUM enables them to exclude this synthetic data from future training sets, thereby preserving model stability.
>
> Finally, to support accountability, our framework naturally supports tracing the generated content back to the source model and user, as demonstrated in our experiments.

---

> ### Author Response · Authors · 2025-11-21
> **Answer to comments. Part 5 - Technical Clarifications**
>
> >**Based on the simplicity illustrated in Figure 1, can the authors elaborate on the specific technical challenges addressed by their approach? In what ways does the method go beyond the obvious, and what unique difficulties does it solve?**
>
> We thank the Reviewer for this question. While the final architecture of SERUM is intentionally designed to be elegant and efficient, arriving at this solution required overcoming several non-trivial technical problems and challenging established assumptions in the field. The unique aspect of SERUM is that it simplifies and unifies the strengths of prior approaches, specifically by avoiding expensive diffusion inversion in favor of a lightweight external detector.
>
> A primary technical challenge was overcoming detector overconfidence, or output binarization (Appendix D). We discovered that a naive implementation of a latent-space detector fails to generalize because it rapidly becomes overconfident on training samples. To solve this, we had to design a specific loss formulation that includes regularization terms for transformed samples. This non-obvious engineering step was critical to prevent overfitting and unlock the high robustness metrics we report, distinguishing our method from simple binary classifiers that fail under attack.
>
> Furthermore, our method fundamentally challenges the prevailing assumption that diffusion inversion is necessary for robust detection. Prior state-of-the-art methods, such as RingID and GaussMarker, rely on the intuition that because the watermark is embedded in the initial noise, detection requires the computationally expensive step of reversing the diffusion process to retrieve that noise. We disprove this barrier by demonstrating that the watermark signal remains algorithmically detectable in the high-dimensional latent space of the generated image, enabling the use of a simple CNN and rendering inversion unnecessary. Proving that one can bridge the gap between the injection space (noise) and detection space (latents) without inversion, and do so with nearly 100% TPR @ 1% FPR across all tested perturbations, is a significant conceptual leap that simplifies the pipeline while drastically boosting speed.
>
> Additionally, SERUM addresses unique difficulties regarding watermark persistence. It is the first diffusion watermark demonstrated to be radioactive, meaning it survives model fine-tuning (including LoRA). This capability requires the signal to persist not just through image perturbations, but through fundamental changes to the model weights themselves.
>
> Finally, we provide a proof that our specific noise injection achieves a provably lower KL divergence to the standard Gaussian distribution than the GaussMarker method, solving the difficulty of balancing detection strength with high image fidelity on a theoretical level.
>
>
> >**Have the authors compared SERUM to non-watermarking detection baselines? How does the method perform when only a simple classifier or traditional watermark is employed instead?**
>
> We thank the Reviewer for this question. We focused our comparison on state-of-the-art generative watermarking methods (GaussMarker [1], RingID [2], Stable Signature [3]) as they represent the current upper bound for robustness in this domain. Additionally, we have conducted further experiments using TrustMark [4] and included these new results in Tables 1 and 13 of our updated submission.
>
> **Regarding non-watermarking baselines (passive detection):** Our comparison relies on the hierarchy established in recent literature. For instance, GaussMarker [1] explicitly demonstrated significant superiority over LatentTracer [5], a leading non-watermarking baseline. Since SERUM consistently outperforms GaussMarker [1] (Table 1 and 2), our results indicate that SERUM surpasses passive detection methods by transitivity. Furthermore, passive methods typically rely on detecting model artifacts, which are fragile and easily destroyed by the perturbations we evaluated (e.g., Gaussian blur, resizing), whereas SERUM relies on an actively injected, robust signal.
>
> **Regarding traditional watermarks:** As discussed in our Introduction and Related Work, traditional frequency-domain watermarks generally struggle to survive generative attacks (such as VAE regeneration or diffusion-based removal) or incur higher quality degradation to achieve comparable robustness. Our results show SERUM achieving >99% TPR on VAE attacks and ~90% on even the most challenging Regen-30 and Rinse-2x25 attacks, scenarios where traditional watermarks fail.

---

> ### Author Response · Authors · 2025-11-21
> **Answer to comments. Part 6 - Significance & References**
>
> >**Does the paper’s contribution fundamentally change our understanding of the detection/provenance problem for AI-generated images, or is it an incremental improvement in watermarking as a technology?**
>
> We believe our work represents a fundamental change rather than an incremental improvement, specifically in how it redefines the feasibility and scope of diffusion model watermarking in two key areas.
>
> First, our work fundamentally changes the approach to watermarking diffusion models. Prior work very heavily relied on DDIM inversion, which has been a major limitation of these methods, as detecting a single watermark incurred a significant computational overhead. We showed that DDIM inversion is not needed, effectively alleviating this problem and shifting from inversion-based to inference-based detection. Furthermore, our work proposes simple latent addition as an effective way to inject watermarks without reducing the diversity of generated images as much as previous approaches. While methods like Tree-Ring [7] and RingID [2] relied on injections in the Fourier space, and GaussMarker [1] enforces specific initial generation noise signs, it is now clear that a simple addition is sufficient when combined with a trained watermark detector.
>
> Second, to the best of our knowledge, we are the first to demonstrate a radioactive watermark for diffusion models that persists even when new model versions are trained on generated data. This represents a fundamental advancement toward breaking data loops and mitigating model collapse - a critical capability that previous diffusion watermarks did not possess.
>
> ---
>
> **References:**
>
> [1] “GaussMarker: Robust Dual-Domain Watermark for Diffusion Models” Kecen Li, Zhicong Huang, Xinwen Hou, Cheng Hong. ICML 2025.
>
> [2] “RingID: Rethinking Tree-Ring Watermarking for Enhanced Multi-Key Identification” Hai Ci, Pei Yang, Yiren Song, Mike Zheng Shou. ECCV 2024.
>
> [3] “The Stable Signature: Rooting Watermarks in Latent Diffusion Models” Pierre Fernandez, Guillaume Couairon, Hervé Jégou, Matthijs Douze, Teddy Furon. ICCV 2023.
>
> [4] “TrustMark: Robust Watermarking and Watermark Removal for Arbitrary Resolution Images” Tu Bui, Shruti Agarwal, John Collomosse. ICCV 2025.
>
> [5] “How to Trace Latent Generative Model Generated Images without Artificial Watermark?” Zhenting Wang, Vikash Sehwag, Chen Chen, Lingjuan Lyu, Dimitris N. Metaxas, Shiqing Ma. ICML 2024.
>
> [6] “Distributions of Angles in Random Packing on Spheres.” . Tony Cai, Jianqing Fan, and Tiefeng Jiang. JMLR 2015.
>
> [7] “Tree-Ring Watermarks: Fingerprints for Diffusion Images that are Invisible and Robust” Yuxin Wen, John Kirchenbauer, Jonas Geiping, Tom Goldstein. NeurIPS 2023.
>
>
> ---
> We would greatly appreciate updating the rating if the above responses address the Reviewer's concerns.

---

> > ### Comment · Reviewer_fqMS · 2025-11-26
> > **Thanks for addressing with great details!**
> >
> > Thanks to the author for addressing my concern in great detail. I will consider increasing my rating accordingly.

---

> ### Author Response · Authors · 2025-11-27
> **Thank You for Your Engagement and Positive Feedback**
>
> Thank you for engaging with us in this discussion. We appreciate your thoughtful feedback and are glad that our detailed responses have been recognized.
>
> We kindly ask you to champion our work and increase the score. We have demonstrated that our SERUM watermark not only scales to thousands of users but can even handle millions. Importantly, we hope our work will inspire the ICLR community to move beyond watermarking and address the larger challenge of identifying generative models, individual users, and eventually providing a comprehensive provenance framework.
>
> If you have any further questions or need additional clarification, we would be happy to help.
>
> Thank you again for your consideration.

---

> > ### Author Response · Authors · 2025-11-27
> > **Thank You for Championing Our Paper**
> >
> > We greatly appreciate the reviewer’s support for our work. Thank you very much for championing our paper.

---

> ### Author Response · Authors · 2025-12-03
> **Update: Large-Scale Multi-User Evaluation & Ablation Study**
>
> We thank the Reviewer again for their constructive feedback and increasing the rating. Following up on our previous response, we have conducted additional large-scale experiments to empirically validate the scalability of our combinatorial watermarking scheme and provided the requested ablation regarding hyperparameter choices (specifically the augmentation sampler's batch size).
>
> We provide these new results to further address Weaknesses 1 and 3:
>
> >**Weakness 1: Although the detector is described as lightweight, performance scaling to much larger numbers of unique watermarks (multi-user setting) may face practical bottlenecks. The paper assesses up to 10 users. In a real-world application, it will be hundreds and even thousands of users. I suggest that the author should invest further in a large-scale application if possible.**
>
> In our initial response, we outlined a combinatorial technique that allows SERUM to scale quadratically. To demonstrate this capability in a realistic large-scale setting, we have now completed an evaluation with **9,045 unique users** (using $m=135$ detectors and subset size $k=2$).
>
> The results, presented in the table below (and in Figure 3 in the updated submission), empirically validate our approach at scale. SERUM maintains exceptional performance even with nearly 10,000 users, achieving a TPR on clean images of **99.96%** and User Identification Accuracy of **99.79%**. Even under difficult geometric perturbations like Crop & Scale, we maintain detection rates of nearly 90%.
>
> **Performance with 9,045 Users**
>
> | Augmentation | TPR @ 1% FPR (%) | User Accuracy (%) |
> | :--- | :---: | :---: |
> | **Clean** | 99.96 | 99.79 |
> | **Rotate** | 78.12 | 82.56 |
> | **JPEG** | 99.30 | 98.58 |
> | **C&S** | 88.70 | 90.29 |
> | **R. Drop** | 97.44 | 96.70 |
> | **Blur** | 99.76 | 99.45 |
> | **S. Noise** | 99.68 | 99.10 |
> | **G. Noise** | 95.24 | 93.42 |
> | **Bright** | 95.76 | 95.18 |
>
>
> Please refer to Section 4.6 and Figure 3 in the updated paper for a comprehensive visualization of these results.
>
> >**Weakness 3: Although architectural details and loss functions are provided, some choices (e.g., watermark noise hyperparameters, batch size, augmentation prioritization) could benefit from clearer justification or ablation.**
>
> In addition to the component analysis provided in our previous response, we have conducted an ablation study on the **augmentation sampler’s batch size ($m$)** to justify our hyperparameter selection.
>
> The results are presented in the table below (and in Appendix G.6, Table 11 of the updated paper). We observe a clear trade-off between robustness and training efficiency. Larger batch sizes significantly increase training time because, at each step, the model must decode latents, apply transformations, and re-encode them. However, larger batches generally improve robustness against hard perturbations by exposing the model to a more diverse set of distortions per update.
>
> Based on this analysis, we selected **$m = 4$** as the optimal balance, achieving near-perfect robustness (99.75% average TPR) while maintaining a manageable training time of under 10 hours.
>
> **Ablation of the augmentation sampler batch size ($m$)**
>
> | Batch Size | Time | Average | Clean | Rotate | JPEG | C&S | R. Drop | Blur | S. Noise | G. Noise | Bright |
> | :--- | :---: | :---: | :---: | :---: | :---: | :---: | :---: | :---: | :---: | :---: | :---: |
> | **precompute** | 2.63 h | 97.16 | 100.00 | 79.72 | 99.90 | 97.18 | 99.56 | 99.96 | 99.86 | 99.18 | 99.06 |
> | **$m = 2$** | 5.66 h | 99.52 | 100.00 | 99.22 | 99.84 | 99.00 | 99.60 | 99.96 | 99.92 | 99.08 | 99.04 |
> | **$m = 4$** | 9.48 h | 99.75 | 100.00 | 99.34 | 99.98 | 99.54 | 99.86 | 100.00 | 100.00 | 99.30 | 99.72 |
> | **$m = 8$** | 15.69 h | 99.80 | 100.00 | 99.74 | 99.90 | 99.62 | 99.80 | 100.00 | 99.94 | 99.50 | 99.74 |
> | **$m = 16$** | 28.15 h | 99.83 | 100.00 | 99.76 | 99.88 | 99.70 | 99.92 | 99.96 | 99.98 | 99.60 | 99.70 |
> | **$m = 32$** | 59.35 h | 99.82 | 100.00 | 99.42 | 99.96 | 99.68 | 99.96 | 100.00 | 99.96 | 99.64 | 99.80 |

---

### Official Review · Reviewer_9VoN · 2025-11-01

**Soundness:** 3
**Presentation:** 3
**Contribution:** 3
**Rating:** 6
**Confidence:** 4

**Summary:**

- This paper proposes a new watermarking framework for DMs that is simpler, more efficient and more robust than existing DM methods.
- Watermarking: A fixed user specific pattern of Gaussian noise is combined (additively) with the initial random noise vector before the diffusion generation process begins. The authors also provide a theoretical guarantee of low KL divergence between the distributions of watermarked and non-watermarked noises due to watermark injection to explain the high output image fidelity using this method
- Detection: A lightweight external detector, which is a CNN is trained to identify the signature of this watermark directly in the VAE latent space of the generated image. Therefore this avoids the need for expensive diffusion inversion for detection which is used in prior tuning free works
- Results:
   1. The method achieves SOTA robustness against a wide variety of image perturbations and advanced generative watermark removal attacks
  2. The method is efficient - it has fast injection and detection times and low training overhead with a low impact on the perceptual quality of the generated images
  3. The method is practical - the authors demonstrates the practical utility through its support for multi user scenarios
  4. The method is shown to have a high degree of 'radioactivity' - the watermark remains detectable even in models that have been finetuned on watermarked data

**Strengths:**

- The main idea proposed in the paper is simple but effective. By decoupling the watermark embedding in the initial noise from the detection mechanism which is a separate and lightweight classifier, the method avoids the bottlenecks of prior works - DDIM inversion which is slow, and expensive finetuning
- The authors have presented sufficient proof to validate better robustness of this method in Table 1 against 3 other DM based methods on all standard perturbations. In addition to the method achieving a high average TPR and low FPR across the 8 perturbations, it also shows consistently high performance across perturbations in comparison to the other methods that have failures on certain transformations ex: RingID on Crop+Scale which falls to 5-6% TPR and Stable Signature on Gaussian Noise which falls to 48-52% TPR
- The method shows superior performance against advanced removal attacks in Table 2 maintaining high TPRs against all baselines and its performance does not collapses under more aggressive attacks like Regen-30 unlike the other methods
- The explanation of generalization in Appendix E and the empirical demonstration that the DDIM inversion process used by the other models actually amplifies the distortions caused by perturbations, making the watermark signal harder to recover is compelling. The method's inversion free, latent space detection is therefore more robust by design
- The method is practical for real world deployment. Table 8 shows that SERUM's detection is orders of magnitude faster than inversion based methods ex: 2.5 min vs 117-140 min to check 5000 images. Watermark injection is 17 ms, and the total detector training time ~9 hrs, including data generation is a fraction of that required for model fine tuning in Stable Signature which is ~57 hours.
- SERUM has a negligible impact on image quality, as confirmed by FID and CLIP scores in Table 4 that are very close to clean generation, and by qualitative examples in Figures 2 and 8

**Weaknesses:**

- Multi user support involves training a unique detector for each user's unique noise pattern. Figure 3 demonstrates that this approach works well for up to 10 users, but this does not guarantee its viability at a large scale. A system with millions of users would require managing millions of individual detector models. As the number of unique noise patterns grows, the probability of 'collisions' where a pattern coincidentally produces a signature detectable by another user's detector may increase. The paper does not provide an analysis of cross detection scores to quantify this risk.
- The evaluation of radioactivity results is confined to a full model fine-tuning scenario. It does not investigate the watermark's persistence under more modern and widely used PEFT methods, such as LoRA.

**Questions:**

- Can the authors comment on the theoretical/empirical limits of one detector per user approach? Have they analyzed the cross detection score distributions to assess the probability of false user identification as the number of users grows into the thousands or millions?
- Can the authors explain on whether the SERUM watermark would be expected to survive parameter-efficient finetuning methods like LoRA where the majority of the base model's weights remain frozen?
- The paper mentions using an augmentation sampler inspired by Prioritized Experience Replay to focus training on more difficult transformations (Appendix A). Can the authors provide an ablation study/further analysis on the impact of this adaptive sampling strategy? How much does it contribute to the final robustness metrics compared to a simpler uniform sampling of augmentations during detector training?

---

> ### Author Response · Authors · 2025-11-21
> **Answer to comments. Part 1 - Multi-User Support**
>
> We thank the Reviewer for their detailed comments. We are pleased that our method is recognized as efficient, practical, and highly radioactive. We also appreciate the positive feedback regarding the clarity and effectiveness of our approach.
>
> In summary, we have significantly enhanced our work to address the raised concerns. Specifically, we:
> 1. Introduced a new combinatorial watermarking scheme that solves the multi-user scalability bottleneck, allowing our method to scale efficiently to millions of users.
> 2. Demonstrated that SERUM remains radioactive even when models are adapted using LoRA.
> 3. Provided comprehensive ablation studies analyzing the impact of the adaptive augmentation sampler and loss components.
>
> We provide detailed answers to all comments and questions below:
>
> >**Question 1: Can the authors comment on the theoretical/empirical limits of one detector per user approach? Have they analyzed the cross detection score distributions to assess the probability of false user identification as the number of users grows into the thousands or millions?**
>
> >**Weakness 1: Multi user support involves training a unique detector for each user's unique noise pattern. Figure 3 demonstrates that this approach works well for up to 10 users, but this does not guarantee its viability at a large scale. A system with millions of users would require managing millions of individual detector models. As the number of unique noise patterns grows, the probability of 'collisions' where a pattern coincidentally produces a signature detectable by another user's detector may increase. The paper does not provide an analysis of cross detection scores to quantify this risk.**
>
> We acknowledge the reviewer's valid concern regarding the logistical challenge of training one detector per user at a scale of millions. To address this, we have enhanced our multi-user protocol by introducing a combinatorial watermarking scheme that significantly reduces complexity and improves scalability.
>
> **Improved Scalability (Quadratic vs. Linear)**: Instead of a 1-to-1 mapping, we now assign each user a unique pair of watermark noise vectors. By training $N$ detectors, we can uniquely identify $\binom{N}{2}$ users. This allows the effective number of users to scale quadratically with the number of detectors. For instance, supporting 1 million users would require managing only $\sim1,415$ detectors, which is computationally feasible and effectively solves the "millions of models" bottleneck.
>
> **Methodology & Robustness**: In this updated approach, we inject two distinct watermarks per image (each with $\alpha = 0.3$, resulting in an effective combined strength of $\alpha = 0.6$). This ensures that the signal remains strong without degrading image quality. The extremely low probability of false positives (collisions) is grounded in the geometry of high-dimensional latent spaces ($d \approx 16,000$). As discussed by Cai et al. [3], random vectors in such spaces are quasi-orthogonal. This implies that the watermark pattern of one user acts as uncorrelated noise to the detectors of other users, effectively minimizing cross-talk and preventing false identification even at a scale of millions. For a detailed description of this updated method and its implementation, please refer to Section 3 and 4.5 in the revised manuscript.
>
> **Empirical Validation**: We validated this approach experimentally with up to 231 users (using just 22 detectors). As shown in the table below, the method maintains high user accuracy across perturbations, comparable to the single-user setting.
>
> **Scalability Analysis (User accuracy)**
> | Augmentation | 55 Users | 105 Users | 153 Users | 231 Users  |
> | :--- | :---: | :---: | :---: | :---: |
> | **Clean** | 99.80% | 99.90% | 99.95% | 99.80% |
> | **Rotate** | 93.29% | 92.74% | 91.54% | 91.29% |
> | **JPEG** | 99.40% | 99.60% | 99.40% | 99.60% |
> | **C&S** | 96.55% | 95.30% | 95.90% | 95.50% |
> | **R. Drop** | 98.90% | 99.20% | 98.80% | 98.45% |
> | **Blur** | 99.80% | 99.85% | 99.75% | 99.70% |
> | **S. Noise** | 99.55% | 99.60% | 99.65% | 99.65% |
> | **G. Noise** | 97.90% | 97.30% | 97.60% | 96.95% |
> | **Bright** | 98.10% | 97.25% | 97.95% | 97.70% |
>
> Additionally, in the updated Appendix K, we provide a theoretical analysis demonstrating that robustness is preserved even when scaling to 10,000+ users.

---

> ### Author Response · Authors · 2025-11-21
> **Answer to comments. Part 2 - LoRA**
>
> >**Question 2: Can the authors explain on whether the SERUM watermark would be expected to survive parameter-efficient finetuning methods like LoRA where the majority of the base model's weights remain frozen?**
>
> >**Weakness 2: The evaluation of radioactivity results is confined to a full model fine-tuning scenario. It does not investigate the watermark's persistence under more modern and widely used PEFT methods, such as LoRA.**
>
> We thank the reviewer for suggesting the evaluation of radioactivity using Parameter-Efficient Fine-Tuning (PEFT). We agree that this is crucial for assessing real-world robustness. We conducted an additional experiment to assess the resilience of SERUM following LoRA (Low-Rank Adaptation) [1] adaptation of Stable Diffusion 1.4.
>
> The results, presented in the updated Table 3 below, demonstrate that the watermark remains detectable with a TPR of 52.30% at 1% FPR. While the detection rate is lower compared to full fine-tuning (77.12%), the watermark signal persists significantly above the random baseline (1%), confirming that SERUM is radioactive even under parameter-efficient adaptation strategies.
>
> **Persistence of SERUM watermark under full fine-tuning vs. LoRA adaptation.**
> | **Model**  | **Adaptation** | **TPR @ 1% FPR** |
> |-|-|-|
> | SD 1.4 | Full Fine-Tuning| 77.12% |
> | SD 1.4 | LoRA | 52.30% |
>
>
> Furthermore, we would like to highlight two critical aspects of these results:
> 1. To the best of our knowledge, SERUM is the first method to demonstrate radioactivity in diffusion models. Recent benchmarks, such as Dubiński et al. [2], indicate that prior diffusion watermarking methods failed to exhibit this property.
> 2. We adopt a strict evaluation protocol: We report TPR at the checkpoint with the lowest evaluation loss on unseen images. This ensures we measure the genuine persistence of the watermark signal (generalization) rather than the model simply overfitting to the watermarked training data.

---

> ### Author Response · Authors · 2025-11-21
> **Answer to comments. Part 3 - Prioritized Experience Replay**
>
> >**Question 3: The paper mentions using an augmentation sampler inspired by Prioritized Experience Replay to focus training on more difficult transformations (Appendix A). Can the authors provide an ablation study/further analysis on the impact of this adaptive sampling strategy? How much does it contribute to the final robustness metrics compared to a simpler uniform sampling of augmentations during detector training?**
>
> We provide the requested ablation studies in the updated version of our paper (Section 4.6). Specifically, we examine the effect of each loss term on clean images, as well as on precomputed and transformed samples. We use the following notation:
> - clean: Only clean images are used in the training (using loss terms $\mathcal{L}_w$ and $\mathcal{L}_n$).
> - precompute: Only clean and precomputed augmentations are applied.
> - transform: We use all loss terms with different values of the parameter $\tau$, which is the temperature term for the augmentation sampler. For $\tau = 0$, the sampler always selects the perturbation with maximal priority, while for $\tau = \infty$ it samples all perturbations uniformly.
>
> The results, presented in Table 6 below, lead to several key observations:
> 1. Our adaptive strategy with $\tau=1.0$ achieves an average TPR of 99.75%, outperforming the uniform sampling baseline ($\tau=\infty$) which scores 99.34%. While the percentage difference appears small, the adaptive sampler significantly improves robustness on the most challenging perturbations. For example, Rotate improves from 95.88% to 99.34%, effectively closing the gap on failure cases.
> 2. Optimal Temperature Balance: We observe that performance peaks at $\tau=1$.
>      - Low Temperature ($\tau \to 0$): The sampler acts too greedily, fixating on the single perturbation with the highest priority until the model overcomes it, before jumping to the next. This lack of batch diversity causes the model to overfit to specific distortions sequentially rather than learning robust features globally.
>    - High Temperature ($\tau \to \infty$): The sampling approaches a uniform distribution. This is suboptimal because the model wastes training capacity on easier augmentations that are already sufficiently covered by the precomputed loss term ($\mathcal{L}_{t}$). The model needs to encounter the hard tail of the distribution more frequently to master difficult attacks.
>    - Optimal Temperature ($\tau = 1$): This setting ensures the detector encounters harder perturbations significantly more frequently, allowing it to learn them effectively, while maintaining enough diversity to avoid overfitting to a single type of noise.
>
> **Impact of watermark components on detection robustness (TPR @ 1% FPR)**
> | **Setup** | **Average** | **Clean** | **Rotate** | **JPEG** | **C&S** | **R. Drop** | **Blur** | **S. Noise** | **G. Noise** | **Bright** |
> | :--- | :---: | :-: | :-: | :---: | :---: | :---: | :---: | :---: | :---: | :---: |
> | **clean** | 78.68 | 100.00 | 14.94 | 99.86 | 59.80 | 77.44 | 100.00 | 98.52 | 66.86 | 88.74 |
> | **precompute** | 97.16 | 100.00 | 79.72 | 99.90 | 97.18 | 99.56 | 99.96 | 99.86 | 99.18 | 99.06 |
> | **transform** ($\tau=0$) | 99.48 | 100.00 | 98.06 | 99.70 | 99.24 | 99.82 | 99.98 | 99.92 | 98.86 | 99.30 |
> | **transform** ($\tau=0.1$) | 99.60 | 100.00 | 99.06 | 99.88 | 99.12 | 99.68 | 100.00 | 99.94 | 99.28 | 99.42 |
> | **transform** ($\tau=0.3$) | 99.64 | 100.00 | 99.00 | 99.94 | 99.38 | 99.66 | 99.96 | 99.94 | 99.44 | 99.48 |
> | **transform** ($\tau=0.5$) | 99.70 | 100.00 | 99.30 | 99.86 | 99.48 | 99.86 | 100.00 | 99.94 | 99.50 | 99.36 |
> | **transform** ($\tau=1.0$) | 99.75 | 100.00 | 99.34 | 99.98 | 99.54 | 99.86 | 100.00 | 100.00 | 99.30 | 99.72 |
> | **transform** ($\tau=2.0$) | 99.67 | 100.00 | 98.34 | 99.90 | 99.56 | 99.82 | 100.00 | 99.98 | 99.70 | 99.76 |
> | **transform** ($\tau=\infty$) | 99.34 | 100.00 | 95.88 | 99.90 | 99.36 | 99.84 | 99.98 | 99.98 | 99.58 | 99.52 |
>
>
> **References:**
>
> [1] “LoRA: Low-Rank Adaptation of Large Language Models” Edward J. Hu, Yelong Shen, Phillip Wallis, Zeyuan Allen-Zhu, Yuanzhi Li, Shean Wang, Lu Wang, Weizhu Chen. ICLR 2022.
>
> [2] “Are Watermarks For Diffusion Models Radioactive?” Jan Dubiński, Michel Meintz, Franziska Boenisch, Adam Dziedzic. ICLR 2025 Workshop WMARK.
>
> [3] “Distributions of Angles in Random Packing on Spheres.” . Tony Cai, Jianqing Fan, and Tiefeng Jiang. JMLR 2015.
>
> ---
> We would greatly appreciate updating the rating if the above responses address the Reviewer's concerns.

---

> ### Comment · Reviewer_9VoN · 2025-11-26
>
> Thank you to the authors for resolving my concerns and questions. I will maintain my current positive score.

---

> > ### Author Response · Authors · 2025-11-27
> > **Thank You for Your Engagement and Positive Score**
> >
> > We are pleased that your concerns and questions were addressed. Thank you for your positive score, this means a lot to us. We would be really grateful for further supporting our work. Specifically, we demonstrated that our SERUM scales efficiently to millions of users and remains radioactive even when new models are adapted on watermarked data using LoRA. We also provided comprehensive ablation studies. We would like this work to contribute to the ICLR community and would be really grateful for further increasing the rating.

---

> ### Author Response · Authors · 2025-12-03
> **Update: Large-Scale Multi-User Evaluation**
>
> We thank the Reviewer again for their constructive feedback and for maintaining their positive score. Following up on our previous responses, we have finalized the large-scale experiments to empirically validate the scalability of our combinatorial scheme for multi-user watermarking.
>
> We provide these new results to further address Weakness 1 and Question 1:
>
> >**Weakness 1: Multi user support involves training a unique detector for each user's unique noise pattern. Figure 3 demonstrates that this approach works well for up to 10 users, but this does not guarantee its viability at a large scale. A system with millions of users would require managing millions of individual detector models. As the number of unique noise patterns grows, the probability of 'collisions' where a pattern coincidentally produces a signature detectable by another user's detector may increase. The paper does not provide an analysis of cross detection scores to quantify this risk.**
>
> > **Question 1: Can the authors comment on the theoretical/empirical limits of one detector per user approach? Have they analyzed the cross detection score distributions to assess the probability of false user identification as the number of users grows into the thousands or millions?**
>
> As outlined in our initial response, we proposed a combinatorial technique that allows SERUM to scale quadratically, solving the logistical bottleneck of the "one detector per user" approach. To demonstrate this capability in a realistic large-scale setting, we have now completed an evaluation with **9,045 unique users** (using $m=135$ detectors and subset size $k=2$).
>
> The results, presented in the table below (and in Figure 3 in the updated submission), empirically validate that SERUM handles real-world scale without degradation. We achieve a TPR on clean images of **99.96%** and a User Identification Accuracy of **99.79%**. Even under difficult geometric perturbations (Crop & Scale), we maintain ~90% accuracy.
>
> **Performance with 9,045 Users**
>
> | Augmentation | TPR @ 1% FPR (%) | User Accuracy (%) |
> | :--- | :---: | :---: |
> | **Clean** | 99.96 | 99.79 |
> | **Rotate** | 78.12 | 82.56 |
> | **JPEG** | 99.30 | 98.58 |
> | **C&S** | 88.70 | 90.29 |
> | **R. Drop** | 97.44 | 96.70 |
> | **Blur** | 99.76 | 99.45 |
> | **S. Noise** | 99.68 | 99.10 |
> | **G. Noise** | 95.24 | 93.42 |
> | **Bright** | 95.76 | 95.18 |
>
> Please refer to Section 4.6 and Figure 3 in the updated submission for a comprehensive visualization of these results.

---

### Author Response · Authors · 2025-11-21
**General Response**

We sincerely thank the reviewers for their thoughtful and constructive feedback. Your comments have helped us significantly strengthen the work. We are encouraged that reviewers found our method novel (Reviewers 9VoN, fqMS), robust (Reviewers 9VoN, fqMS, cDi8), and efficient (Reviewers 9VoN, fqMS, cDi8).

In response to your suggestions, we have made several substantial updates and additions to the submission:
1. We optimized the multi-user scenario, drastically reducing training overhead and ensuring scalability (the number of users scales quadratically with the number of detectors).
2. We demonstrated that our watermark is radioactive when new models are adapted on our watermarked data using LoRA, as requested (Reviewer 9VoN).
3. We performed new experiments exploring different training configurations, including removing augmentation-related loss terms and Prioritized Experience Replay, as well as varying Replay’s temperature hyperparameter (Reviewers 9VoN, fqMS, cDi8).
4. We assessed SERUM and baseline methods under the TPR at 0% FPR metric (Reviewer Emwh), showing that our method remains effective (high TPR) even at this strictest requirement.
5. We evaluated Crop & Scale down to 25% retention (Reviewer Emwh), showing that SERUM maintains high robustness where Stable Signature collapses.
6. We added comparisons against TrustMark, a state-of-the-art post-processing watermarking method (Reviewers fqMS, cDi8). The comparison highlights the significant advantages of our approach over both in-processing and post-processing alternatives.
7. We evaluated SERUM using the Diffusion Transformer (DiT) architecture to demonstrate that our method generalizes across different model backbones (Reviewer cDi8).
8. Section titles have been shortened in the updated submission according to the recommendations (Reviewer fqMS).

---

### Author Response · Authors · 2025-12-03
**Update on Additional Experiments**

We thank the Reviewers for their engagement and constructive feedback throughout the rebuttal process. We are particularly grateful for the willingness to increase scores (Reviewers fqMS, cDi8) and for the continued high evaluation (Reviewer 9VoN).

To fully address the remaining suggestions regarding scalability, advanced threats, and hyperparameter justification, we have completed the following additional experiments:

1. **Large-Scale Multi-User Evaluation (Reviewers fqMS, Emwh, 9VoN, cDi8):** We empirically validated our combinatorial watermarking scheme with **9,045 unique users**. The results confirm our theoretical projections, maintaining 99.96% TPR on clean images and ~90% accuracy even under difficult geometric perturbations, demonstrating true scalability.
2. **Advanced Attacks (Reviewer cDi8):** We evaluated SERUM against two highly challenging attacks: **CtrlRegen** and **Image-to-Video** transformations. SERUM achieves the highest robustness among all methods (e.g., 99.64% TPR on CtrlRegen vs. <1% for baselines), proving its superior generalization beyond trained perturbations.
3. **Batch Size Ablation Study (Reviewers fqMS, cDi8):** We conducted a detailed ablation on the augmentation sampler's batch size ($m$). The results justify our selection of $m=4$ as the optimal trade-off, achieving near-perfect robustness (99.75% avg TPR) while keeping training time highly efficient (<10 hours).

We believe these final results fully support the paper's claims and hope they further solidify your positive assessment.

---

### Meta-Review · Area_Chair_JYKh · 2025-12-26

**Summary:**

This paper received initial review ratings of 6, 6, 4 and 4. The reviewers appreciated the robustness, simplicity, and performance of the proposed method and the quality of the presentation. They had concerns about scalability and raised questions about crop robustness, evaluation schemes and metrics, and the hyperparameter setting. The authors provided detailed explanations, clarification and additional experiments to address reviewers’ questions and concerns. Before the stop of the rebuttal discussion, three reviewers, 9VoN, fqMS and Emwh, whose initial ratings were 6, 6 and 4, respectively, responded to authors. They agreed that the authors addressed their concerns and answered their questions. 9VoN maintained their positive rating and fqMS and cDi8 would like to raise their ratings. The AC checked the authors’ response to reviewer Emwh. The AC believes that the authors well addressed their concerns. Thus, AC recommends accepting this paper.

**Reviewer Concerns:**

The authors addressed all reviewers' concerns.

**Reviewer Scores:**

Before the stop of the rebuttal discussion, three reviewers, 9VoN, fqMS and Emwh, whose initial ratings were 6, 6 and 4, respectively, responded to authors. They agreed that the authors addressed their concerns and answered their questions. 9VoN maintained their positive rating and fqMS and cDi8 would like to raise their ratings.

---

### Decision · Program_Chairs · 2026-01-26

Accept (Poster)